# X-ray structure and enzymatic study of a bacterial NADPH oxidase highlight the activation mechanism of eukaryotic NOX

**Isabelle Petit-Hartlein[1†], Annelise Vermot[1†], Michel Thepaut[1], Anne-Sophie Humm[2], Florine Dupeux[1,2], Jerome Dupuy[1], Vincent Chaptal[3], Jose Antonio Marquez[2,4], Susan ME Smith[5]\*, Franck Fieschi[1,6]\***

[1]Univ. Grenoble Alpes, CNRS, CEA, Institut de Biologie Structurale, Grenoble, France; [2]European Molecular Biology Laboratory, Grenoble, France; [3]CNRS-Lyon 1 University Laboratory 5086, IBCP, Lyon, France; [4]ALPX S.A.S. 71, Grenoble, France; [5]Department of Molecular and Cellular Biology, Kennesaw State University, Kennesaw, United States; [6]Institut Universitaire de France, Paris, France

**\*For correspondence:**
ssmit325@kennesaw.edu (SMES);
franck.fieschi@ibs.fr (FF)

[†]These authors, listed in alphabetical order, contributed equally to this work

**Competing interest:** The authors declare that no competing interests exist.

**Abstract** NADPH oxidases (NOX) are transmembrane proteins, widely spread in eukaryotes and prokaryotes, that produce reactive oxygen species (ROS). Eukaryotes use the ROS products for innate immune defense and signaling in critical (patho)physiological processes. Despite the recent structures of human NOX isoforms, the activation of electron transfer remains incompletely understood. SpNOX, a homolog from *Streptococcus pneumoniae*, can serves as a robust model for exploring electron transfers in the NOX family thanks to its constitutive activity. Crystal structures of SpNOX full-length and dehydrogenase (DH) domain constructs are revealed here. The isolated DH domain acts as a flavin reductase, and both constructs use either NADPH or NADH as substrate. Our findings suggest that hydride transfer from NAD(P)H to FAD is the rate-limiting step in electron transfer. We identify significance of F397 in nicotinamide access to flavin isoalloxazine and confirm flavin binding contributions from both DH and Transmembrane (TM) domains. Comparison with related enzymes suggests that distal access to heme may influence the final electron acceptor, while the relative position of DH and TM does not necessarily correlate with activity, contrary to previous suggestions. It rather suggests requirement of an internal rearrangement, within the DH domain, to switch from a resting to an active state. Thus, SpNOX appears to be a good model of active NOX2, which allows us to propose an explanation for NOX2's requirement for activation.

## eLife assessment

In this manuscript, the authors investigate the properties of prokaryotic NADPH oxidases (NOX) and discuss the implications for NOX regulation and function. The structure of the *S. pneumoniae* Nox protein is an **important** step forward in our understanding of procaryotic NOX enzymes and the characterization and interpretation are **convincing**. The results will be of interest to structural biologists as well as biochemists focusing on enzymatic functions.

## Introduction

NADPH oxidases (NOX) are membranous enzymes that 'professionally' produce the reactive oxygen species (ROS) superoxide or $H_2O_2$ (*Vermot et al., 2021*); the 'professional' production in NOX contrasts with many other sources of cellular ROS which result from metabolic leaking or byproducts. The first-characterized human enzyme of the NOX family, now called NOX2, has been

known since 1978 (*Segal and Jones, 1978*), and provided a paradigm of its ROS product as a cytotoxic agent in defense mechanisms. However, from 1999 to 2001, six other isoforms of NOX were discovered: NOX1/3/4/5 and DUOX1/2 (*Bánfi et al., 2001*; *De Deken et al., 2000*; *Dupuy et al., 1999*; *Geiszt et al., 2000*; *Kikuchi et al., 2000*; *Suh et al., 1999*). These other isoforms of NOX function in many tissues and cell types and the ROS they produce play roles beyond cytotoxic defense, as biochemical and signaling agents. In addition to the identification of NOX2 deficiency as the cause of chronic granulomatous disease (*Segal and Jones, 1978*; *Segal and Jones, 1979*), dysregulation of other NOX family members has since been highlighted in pathophysiologies like hyperthyroidism (*De Deken et al., 2000*; *Dupuy et al., 1999*) and inflammatory bowel diseases along with well-documented participation in normal physiological process like cardiovascular tone regulation (*Rajagopalan et al., 1996*), sense of balance (*Bánfi et al., 2004*), and fertility (*Musset et al., 2012*) among others. Interestingly, these integrated processes directly link to multicellularity leading to the belief that true NOXes are found only in multicellular eukaryotes (*Lalucque and Silar, 2003*).

The various isoforms display significant diversity in assembly with accessory membrane proteins (p22[phox] for NOX1-4, DUOXA1/2 for the corresponding DUOX1/2) or not (NOX5); in phosphorylation-promoted association of cytosolic factors required for activation (NOX 1–3) or not (NOX4-5, DUOX1/2); in the presence of EF hands conferring $Ca^{2+}$ dependence (NOX5, DUOX1/2) or not (NOX1-4); and in the species of ROS produced, superoxide for NOX1-3,5 and $H_2O_2$ for NOX4 and DUOXs enzymes (*Vermot et al., 2021*).

Despite their variety, all NOX family members share a common catalytic core comprised of two conserved domains. The transmembrane (TM) domain belongs to the Ferric Reductase Domain (FRD) superfamily, which consists of six TM helices containing conserved histidines in TM 3 and 5 that chelate up to two heme b cofactors (*Zhang et al., 2013*). The cytosolic dehydrogenase (DH) domain belongs to the Ferredoxin NADP +Reductase (FNR) family which has motifs to bind a flavin cofactor and a pyridine dinucleotide NAD(P)H (*Taylor et al., 1993*). Thus, the NOX core catalytic domain contains the necessary machinery to move electrons from the electron donor – NADPH – through the flavin and heme cofactors to the ultimate electron acceptor, molecular oxygen, to make the ROS product.

The fusion of the TM (N-terminal) and DH (C-terminal) domains constitutes the conserved architecture for the catalytic subunit of the NOX family members. The widespread appearance of separate FRD and DH domains in prokaryotes motivated a search for prokaryotic NOX homologs. First, separate *E. coli* FRD and FNR representatives MsrQ and Fre were identified and shown to interact, producing a NOX-like flow of electrons from NADPH to flavin and through the hemes to an electron acceptor (*Juillan-Binard et al., 2017*). Another bioinformatic search identified a large number of prokaryotic sequences with fused FRD and FNR domains (*Hajjar et al., 2017*). We demonstrated that one such protein from *Streptococcus pneumoniae*, called SpNOX, is a bona fide NOX that produces ROS and shares the spectral signature and biochemical hallmarks (resistance to cyanide, inhibition by DPI) of eukaryotic NOX (*Hajjar et al., 2017*), although its biological function in *S. pneumoniae* has not yet been determined. These results demonstrate definitively that NOXes also exist in prokaryotes. SpNOX purifies as a monomer and is constitutively active (*Hajjar et al., 2017*), so it represents the minimal functional core shared by all NOX family members. Furthermore, unlike eukaryotic NOX family members, it overexpresses well in *E. coli* and retains robust activity in detergent solution. These qualities make SpNox well suited to structural determination by several methods.

Published crystal structures of separate TM and DH domains of a NOX5 homolog CsNOX from the cyanobacterium *Cylindrospermum stagnale* (*Magnani et al., 2017*) provided novel structural information about the NOX family. A low-resolution structure of full-length SpNOX (*Vermot et al., 2020*) provided the first look at the fused TM and DH domains in NOX proteins, while cryo-EM structures of mammalian DUOX1 (*Sun, 2020*; *Wu et al., 2021*) and NOX2 (*Liu et al., 2022*; *Noreng et al., 2022*) furnished important new insights. Still, much remains to be discovered regarding electron transfer, the essential catalytic function of the NOX family. SpNOX thus remains an attractive prototype through which to examine structure/function relationships common to NOX family members.

Here, we obtained X-ray structures of WT and F397W mutant of DH-only, and of a F397W mutant of full-length SpNOX. We combined enzymological and structural characterization to elucidate insights into substrate and cofactor recognition, as well as kinetic and mechanistic aspects of electron transfer for the NOX enzyme family. Comparative analysis of this constitutively active prokaryotic NOX

with the previously described human NOX2 and DUOX1 structures provides a molecular explanation NOX2's requirement for an activation step.

## Results and discussion

### Production of SpNOX and SpNOX$_{DH}$ domain

SpNOX shares only about 25% identity with the human NOX homologs; nonetheless the NOX family members share several well-conserved motifs identified in eukaryotic NOX by *Massari et al., 2022*. *Figure 1A* shows the logos corresponding to the eukaryotic NOX motifs in *Massari et al., 2022*, but constructed from our alignment containing about half prokaryotic sequences. Other than the motif identified as NADPH3, for which our joint prokaryotic-eukaryotic NOX alignment lacked consensus, SpNOX (and other prokaryotic NOX) recognizably share all the motifs identified by *Massari et al., 2022*.

The C-terminal motif identified by our alignment includes several residues past the C-terminal end of the motif identified from eukaryotes, indicating the high frequency of additional residues in prokaryotic NOX. A specific acid/base pair ($E_{283}R_{284}$ human NOX2 numbering) highly conserved in eukaryotic NOX appears directly after the predicted end of the 6$^{th}$ TM helix (*Sharpe et al., 2010*), at the inter-domain hinge (*Figure 1B*); SpNOX has the similar pair $Q_{180}K_{181}$ at this position which prompted us to define K181 as the start of SpNOX$_{DH}$. A typical SDS-PAGE of purified SpNOX and SpNOX$_{DH}$ is shown (*Figure 1—figure supplement 1*).

### Optimization of the SpNOX$_{DH}$ thermostability and activity

SpNOX$_{DH}$ showed a tendency to aggregate, so we investigated buffer conditions to optimize its stability, solubility and activity (*Gekko and Timasheff, 1981*; *Martins de Oliveira et al., 2018*). SpNOX$_{DH}$ thermostability and activity were assessed through wide ranges of salt concentrations (0.05–1 M) and glycerol percentage (0–20%). The temperatures of thermal unfolding were determined by nanoDSF and demonstrated a constant improvement of $T_m$ with NaCl concentration and glycerol percentage increases (*Figure 2A and B*). SpNOX$_{DH}$ activity was assessed by reduction of cytochrome *c*. In reactions without NaCl and glycerol, the DH domain efficiently supported cytochrome *c* reduction (*Figure 2C and D*). In contrast to thermal stability, however, SpNOX$_{DH}$ exhibits a dramatic loss of activity with increasing NaCl and glycerol. We generated a 3D surface plot of Tm as a function of NaCl and glycerol with specific activity as color scale (*Figure 2E*) which allowed us to optimize buffer conditions jointly. The region highlighted by (○) in *Figure 2E* indicates that at 300 mM NaCl and 5% glycerol SpNOX$_{DH}$ exhibits both high activity (5 mol reduced cyt. *c*/s/mol.SpNOX$_{DH}$) and a good thermostability (melting temperature of 60 °C).

We then optimized the pH in order to conjointly maximize the thermostability and the activity of SpNOX$_{DH}$. Direct comparison of the kinetic parameters of SpNOX$_{DH}$ and SpNOX necessitated identifying a pH that would work well for both constructs. The dependences of SpNOX$_{DH}$ activity (*Figure 2—figure supplement 1A*) and stability (*Figure 2—figure supplement 1B*), evaluated along a range of pH (respectively 5–7 and 4–9.5) in the presence of 300 mM NaCl 5% glycerol, exhibit a progressive decrease with increasing alkalinity. Like SpNOX$_{DH}$, the *thermostability* of SpNOX follows a similar and progressive decrease of $T_m$ with alkalinity (*Figure 2—figure supplement 1B*), but in contrast the *activity* of SpNOX shows an optimum at higher alkalinity (*Figure 2—figure supplement 1A*). The two constructs show similar activity at pH 6.5, so we chose this as the pH. Thus, for all experiments in which we compare activity of the full-length SpNOX to the truncated SpNOX$_{DH}$, we chose a standard buffer of 50 mM bis TRIS-propane pH 6.5, 300 mM NaCl, 5% glycerol.

### Functional characterization of SpNOX$_{DH}$

The DH domain of NOXes, including SpNOX, as a homolog of the FNR superfamily, should catalyze the transfer of two electrons from the cytosolic NADPH to the FAD coenzyme; in full-length NOX the electrons are then transferred singly to the hemes and across the membrane to molecular $O_2$. The functionality of the DH domain of SpNOX was evaluated by monitoring the oxidation of NADPH substrate and also by monitoring ferricytochrome c reduction which in the absence of the TM domain indicates the capacity of FAD to transfer electrons to soluble $O_2$. Superoxide generation was assessed in aerobic conditions using SOD-inhibitable ferricytochrome c reduction.

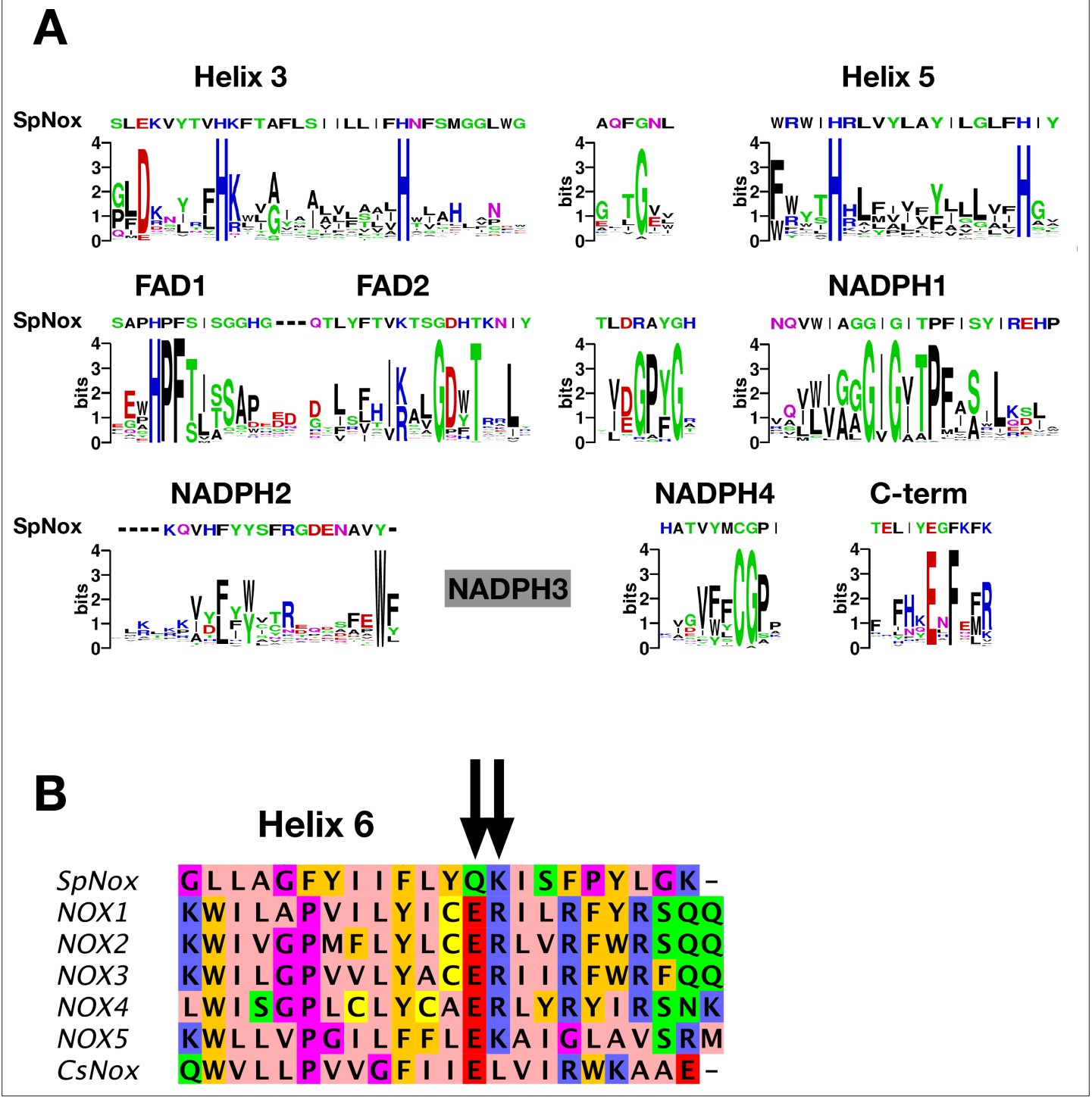

**Figure 1.** NOX-specific sites conserved in SpNOX. (**A**) Alignment of the SpNOX sequence with motifs identified in eukaryotic NOX by Massari et al (61). Alignments of TM and DH domains including prokaryotic and eukaryotic NOX sequences, as described in Materials and methods, were used to construct logos of sub-sequences identified as NOX-specific sites in *Massari et al., 2022*. The NADPH3 motif as defined in *Massari et al., 2022* is no longer relevant when adding prokaryotic sequence in alignments and is not presented here (grey box). (**B**) The region of human NOX1-5, CsNOX, and SpNOX spanning TM helix 6 and into the first beta strand of the DH domain was aligned to identify the putative end of the TM domain and beginning of the DH domain. Arrows indicate the ER ($E_{283}R_{284}$ in NOX2 numbering) pair highly conserved in eukaryotic NOX, and the corresponding $Q_{180}K_{181}$ pair in SpNOX.

The online version of this article includes the following source data and figure supplement(s) for figure 1:

**Figure supplement 1.** SDS PAGE of purified SpNOX and SpNOX_DH.

*Figure 1 continued on next page*

*Figure 1 continued*

**Figure supplement 1—source data 1.** Original file for the SDS-PAGE of SpNOX-full length and SpNOXDH.

Like purified or membrane preparations of both the full-length and DH constructs of other NOX family members (*Han et al., 2001*; *Nisimoto et al., 2004*; *Nisimoto et al., 2010*), SpNOX$_{DH}$ requires supplemental FAD to sustain both superoxide production, which can be observed in the presence of Cyt c (*Figure 3A*), and NADPH oxidation, which can be observed in the absence of Cyt c (*Figure 3B*). The dependence of SpNOX$_{DH}$ on flavin in the oxidation of NADPH was assessed by addition of diphenyleneiodonium (DPI), a common specific flavoenzyme inhibitor (*Nisimoto et al., 2014*), which inhibited the enzyme activity (*Figure 3C*).

In *Figure 3D* NADPH oxidation and superoxide generation by SpNOX$_{DH}$ in aerobic conditions was followed simultaneously at both 340 and 550 nm. The addition of superoxide dismutase (SOD) led to a ~40% inhibition of SpNOX$_{DH}$'s Cyt. *c* reductase activity, confirming that the reduction of Cyt. *c* observed in the system is partly due to $O_2^{\bullet-}$ generation (*Figure 3D* and *Table 1*). However, the remaining~60%Cyt. *c* reductase activity suggests direct electron transfer from the FADH$_2$ produced by the SpNOX$_{DH}$ to the Cyt. *c*. Thus, SpNOX$_{DH}$, separated from the TM domain, retained the redox properties and sensitivity displayed in the context of the whole SpNOX protein (*Hajjar et al., 2017*). The strict requirement of FAD addition for SpNOX$_{DH}$ activity and its µM level of affinity suggests that the flavin behaves as a co-substrate rather than a prosthetic group. As an isolated domain, SpNOX$_{DH}$ may work as a flavin reductase enzyme (*Gaudu et al., 1994*; *Fieschi et al., 1995*; *Nivière et al., 1996*), making an interesting parallel with the *E. coli* flavin reductase *fre* (also in the FNR family) which reduces the MsrQ transmembrane component (in the FRD family, like the NOX TM domain, see Figure 7; *Caux et al., 2021*; *Juillan-Binard et al., 2017*).

## Comparative flavin recognition properties and activities of SpNOX$_{DH}$ vs SpNOX

To characterize the flavin reductase activity of the DH domain and the relative contributions to FAD binding of the TM and DH domains of SpNOX, we determined the $K_m$ and $k_{cat}$ of SpNOX and

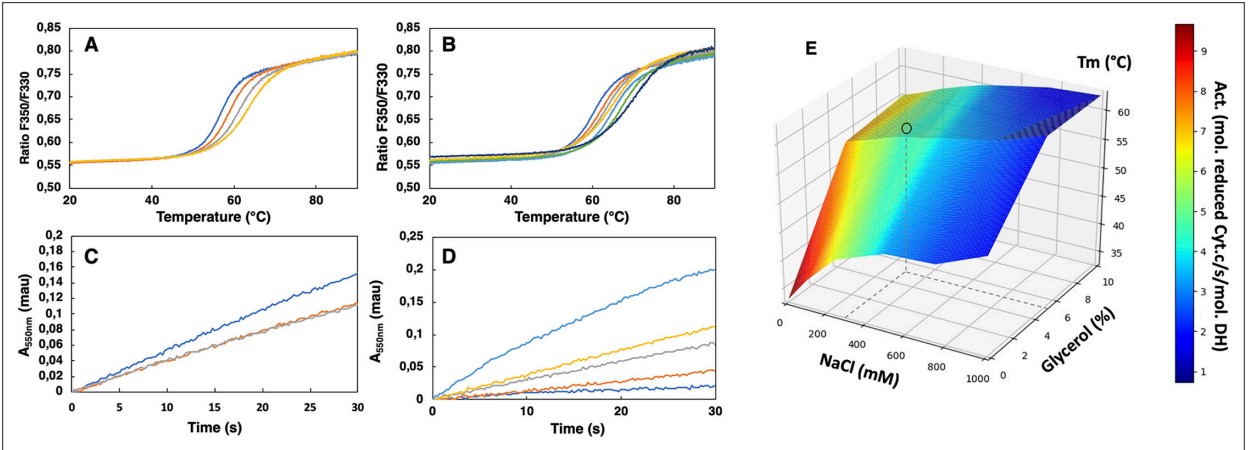

**Figure 2.** Buffer optimization for improvement of SpNOX$_{DH}$ thermostability and activity. (**A**) Thermal unfolding curves of SpNOX$_{DH}$ (0.2 mg.mL$^{-1}$) in 50 mM B bis TRIS-propane buffer pH 6.5, 300 mM NaCl with various glycerol percentages: 5% (dark blue), 10% (orange), 15% (grey), 20% (yellow). (**B**) Thermal unfolding curves of SpNOX$_{DH}$ (0.2 mg.mL$^{-1}$) in bis TRIS-propane buffer pH 6.5, 5% glycerol with various NaCl concentrations: 100 mM (dark blue), 200 mM (orange), 300 mM (grey), 400 mM (yellow), 500 mM (light blue), 750 mM (green), 1M (indigo). (**C**) Cytochrome *c* reduction in presence of SpNOX$_{DH}$ (1 µg) NADPH (200 µM) FAD (10 µM) in the standard 50 mM bis TRIS-propane buffer at pH 6.5, 300 mM NaCl in presence of various glycerol percentages: 0% (blue line), 5% (orange line), 10% (grey line). (**D**) Cytochrome *c* reductase activity with SpNOX$_{DH}$ (1 µg), NADPH (200 µM), FAD (10 µM), glycerol (5%), in presence of various NaCl concentrations: 0 mM (light blue), 100 mM (yellow), 200 mM (grey), 400 mM (orange), 1M (dark blue). (**E**) 3D surface representation of T$_m$ dependence with NaCl concentrations and glycerol percentages. (O) depicts the optimum used for further experiments of this study.

The online version of this article includes the following figure supplement(s) for figure 2:

**Figure supplement 1.** Comparison of pH dependence of SpNOX and SpNOX$_{DH}$ thermostability and activity.

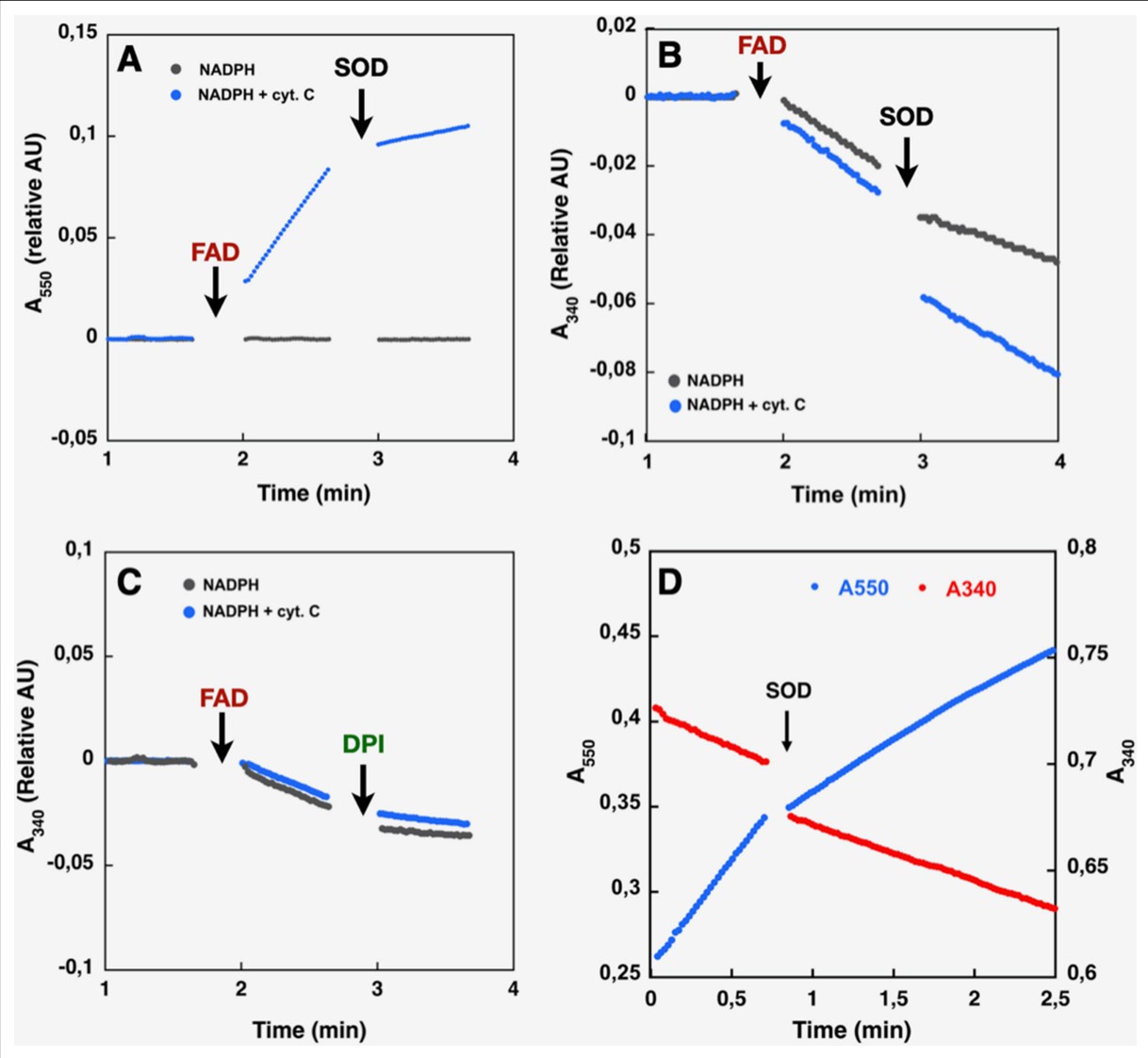

**Figure 3.** Effect of cyt. *c*, FAD, SOD, and DPI supplementation on SpNOX$_{DH}$ NADPH oxidase activity. Assays performed with NADPH (200 µM) and SpNOX$_{DH}$ (1 µg) in the initial mix. Monitoring of cytochrome *c* reduction (Cyt. *c* reductase activity) (**A**) or NADPH oxidation (flavin reductase activity) (**B**) along with successive addition of FAD and SOD (**A** and **B**) or DPI (**C**) with Cyt. c (blue trace) or without (grey trace) in the initial mix. (**D**) Monitoring simultaneously NADPH oxidation (red) and cytochrome c reduction (blue) in presence of SpNOX$_{DH}$, cytochrome *c*, NADPH and FAD, addition of SOD is indicated by an arrow.

SpNOX$_{DH}$ cytochrome *c* reductase activity using flavins FAD, FMN, riboflavin, and lumiflavin (**Table 2**). We observed a progressive loss of affinity for SpNOX as flavin size diminishes, with 263-fold higher affinity for FAD than for lumiflavin (**Table 2**). In contrast, the SpNOX$_{DH}$ retains constant affinity equal to that of lumiflavin (~10 µM) regardless of the flavin analog. These results suggest that the isolated DH domain binds to FAD primarily at the isoalloxazine ring of the cofactor, and the addition of ribitol (riboflavin), phosphate groups (FMN), and adenosine nucleotide (FAD) do not further interact significantly with the isolated DH domain. However, the full-length SpNOX appears to interact along the entire length of the FAD cofactor, indicating significant interactions between the FAD and the TM domain. This FAD binding site shared between the TM and DH domains may contribute to the correct placement of the cofactor for electron transfer to the proximal heme. On the other hand, in full-length SpNOX the $k_{cat}$ appears to be independent of the flavin derivative used, suggesting that the binding interactions between TM and FAD play no catalytic role and that only the isoalloxazine ring needs to be taken into account for the electron pathway from NADPH to the first heme. Finally, the $k_{cat}$ of

**Table 1.** Molecular specific activity monitoring either Cyt. C reduction or NADPH oxidation (flavin reductase activity).

Activities were measured for the different compositions of the initial mixture and addition of the multiple reactants. The sample sizes for the activity measurements were between 2 and 3.

| | Cyt. C reductase activity | Flavin reductase activity |
|---|---|---|
| | mole of cyt. c reduced.s-1.mol-1 SpNOXDH | mole of NADPH oxidized.s-1.mol-1 SpNOXDH |
| FAD | n.a. | 4,76±0.51 |
| FAD +Cyt .C | 6.64±0.51 | 6.34±0.33 |
| FAD +Cyt .C+SOD | 3.97±0.44 | 5.26±1.42 |
| FAD +DPI | n.a | 1.18±1.42 |

full-length and DH-only constructs are the same order of magnitude, with ratios around 1 (*Table 2*), showing that the presence (or absence) of the hemes in the TM domain has little effect on the rate. It follows that the reduction of FAD – that is, hydride transfer from the nicotinamide – is the rate limiting step in the electron transfer from substrate to $O_2$.

## SpNOX uses NADPH and NADH with similar affinity and activity

The typical Michaelis Menten type kinetics parameters of SpNOX and SpNOX$_{DH}$ were obtained for increasing NAD(P)H concentrations (*Table 3*). Full length and DH constructs showed the same order of magnitude affinity to each other for NADPH or NADH as substrate (*Table 3*). Furthermore, the value of 1 for NADPH/NADH affinity for each construct shows that SpNOX does not discriminate significantly

**Table 2.** SpNOX and SpNOX$_{DH}$ affinity for flavins.

$K_m$ and $k_{cat}$ of SpNOX and SpNOX$_{DH}$ for FAD, FMN, riboflavin, and lumiflavin. $K_{mDH}/K_m$ and $k_{catDH}/k_{cat}$ indicate the ratio between $K_m$s or $k_{cat}$s measured on the DH domain and on the full-length SpNOX protein. *Table 2—source data 1* shows one series of the kinetic studies corresponding to the parameters presented here. The sample sizes for the activity measurements were between 2 and 3.

**FAD**  **FMN**  **Riboflavin**  **Lumiflavin**

| | $K_m$ (μM) | | | $k_{cat}$ (s$^{-1}$) | | |
|---|---|---|---|---|---|---|
| | SpNOX | SpNOX$_{DH}$ | $K_{mDH}/K_m$ | SpNOX | SpNOX$_{DH}$ | $k_{catDH}/k_{cat}$ |
| FAD | 0.049±0.01 | 12.89±0,69 | 263 | 4.68±0.74 | 6.73±0.74 | 1.44 |
| FMN | 0.92±0.01 | 7.89±0.48 | 8.6 | 3.01±0.04 | 5.14±0.14 | 1.70 |
| Riboflavin | 8.28±0.01 | 16.76±1.47 | 2 | 7.37±0.38 | 2.77±0.16 | 0.38 |
| Lumiflavin | 16.56±2.77 | 15.85±1.07 | 1 | 4.67±0.45 | 4.14±0.11 | 0.89 |

The online version of this article includes the following source data for table 2:

**Source data 1.** Michaelis Menten analysis of SpNOX and SpNOXDH as a function of the flavin substrate.

**Table 3.** $K_m$ and $K_{cat}$ using NADPH or NADH as electron donor for the SpNOX WT, F399W and F397S full-length constructs or the corresponding versions of the DH domain (SpNOX$_{DH}$). F399W is in italics to indicate that position 399 is not homologous to the terminal aromatic in FNR; the W mutation at the homologous position, F397W, did not have sufficient activity to perform such analysis and thus does not appear in this table. WT NADPH/NADH is the ratio of the $K_m$ or $K_{cat}$ for NADPH vs NADH. For SpNOX$_{DH}$ WT $K_m$s were determined by monitoring both Cyt. c reductase activity (at 550 nm) or flavin reductase activity (at 340 nm) when reported. The kinetic studies corresponding to the parameters presented here are in *Table 3—source data 1*. The sample sizes for the activity measurements were between 2 and 3.

| Substrat | | $K_m$ (µM) | | | $k_{cat}$ (s$^{-1}$) | |
|---|---|---|---|---|---|---|
| | | SpNOX | SpNOX$_{DH}$ | | SpNOX | SpNOX$_{DH}$ |
| | *Assays* | *Cyt. C red* | *Cyt. C red* | *FAD red* | *Cyt. C red* | *Cyt. C red* |
| | WT | 41.5±5 | 106±31 | 87.89±13 | 6.42±0.6 | 5.42±1.15 |
| NADPH | *F399W* | n.d. | 86.7±6.5 | - | n.d. | 9.07±0.5 |
| | F397S | 33.76±5 | 79.3±19.5 | - | 10.80±0.5 | 8.29±1.3 |
| | WT | 41.81±4.4 | 133±21 | 80.27±9.1 | 3.5±0.2 | 4.41±0.5 |
| NADH | *F399W* | n.d. | 148.6±46 | - | n.d. | 8.65±2 |
| | F397S | 25.46±2 | 66.9±10.3 | - | 10.3±0.4 | 6.20±0.5 |
| WT NADPH/NADH | | 1 | 0.8 | 0.96 | 1.8 | 1.2 |

The online version of this article includes the following source data for table 3:

**Source data 1.** Michaelis Menten analysis of wild type and mutant SpNOX and SpNOXDH as a function of the nicotinamide-based electron donor.

between NADPH and NADH; the agreement of two different assays – cytochrome c reduction and NADPH oxidation – strengthens this observation. Thus, in SpNOX, the NAD(P)H binding energy arises from interactions just in the DH domain with little contribution from the TM domain of SpNOX. The lack of discrimination of SpNOX between NADH and NADPH represents a significant difference from eukaryotic NOX which strongly select NADPH as substrate. In these experiments that used a range of NAD(P)H concentration, $k_{cat}$'s of the same order of magnitude for both constructs again supports hydride transfer as the limiting step in NOX electron flow.

## Structural characterization of SpNOX and SpNOX$_{DH}$ : mutation of F397

We obtained large hexagonal crystals of SpNOX but with poor diffraction power beyond 5 Å, and only 6% between 3 and 5 Å, with clear anisotropy. We therefore tried stabilizing mutations based on the work of *Deng et al., 1999* in pea FNR, homologous to the NOX DH domain. They showed that the FNR C-terminal Y308 stabilizes the isoalloxazine site of the FAD cofactor and regulates access of the NADP + substrate to this site; other works extended the discussion of the importance of this residue to the NOX DH domain (*Zhen et al., 1998*; *Kean et al., 2017*). We reasoned that changing the corresponding residue of SpNOX to W would stabilize FAD binding and inhibit NADPH binding, thereby reducing mobility of the two subdomains relative to each other. Supporting this approach, Magnani et al reported the stabilizing effect in CsNOX of the serendipitous insertion of a Trp that stacks with the FAD isoalloxazine ring (*Ceccon et al., 2017*; *Magnani et al., 2017*). Similarly, we reasoned that changing the corresponding F residue to S would increase NADPH access to FAD, potentially facilitating crystals with NADPH bound.

Based on sequence alignment (*Figure 1*), either of two phenylalanine residues in SpNox, F397, and F399, could correspond to the conserved C-terminal residue in FNR. We mutated both positions to tryptophan in the SpNOX$_{DH}$ construct and monitored activity to test their effects. In our standard buffer used for comparing DH with full-length constructs, F399W mutants showed normal activity (*Table 3*), while F397W mutants showed no activity (*Figure 4*). We interpreted this to mean that W in position 397 prevents hydride transfer by immobile stacking with the isoalloxazine ring of FAD, while

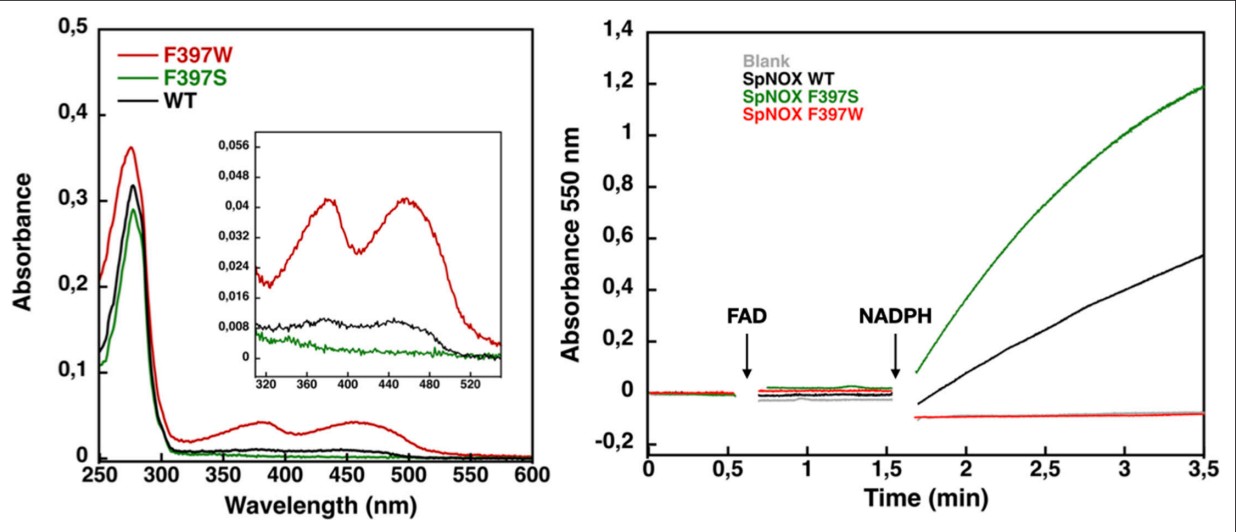

**Figure 4.** Flavin content and activity measurement of full-length SpNOX WT, SpNOX F397S and F397W. (**A**) UV-Visible spectra of SpNOX$_{DH}$ WT, and mutants. Inset zoom on the 320–500 nm window, a specific spectroscopic feature of the oxidized flavin isoalloxazine ring. (**B**) Reduced cytochrome c was monitored at 550 nm for SpNOX WT (black), F397W (red), and F397S (green). A no-protein negative control is shown in grey. See also *Figure 4—figure supplement 1* for spectral characterization of full-length SpNOX mutants.

The online version of this article includes the following figure supplement(s) for figure 4:

**Figure supplement 1.** Characterization of SpNOX, SpNOX-F397W and SpNOX-F397S mutants.

W in position 399 has little effect on hydride transfer. These results indicate that F397 of SpNOX corresponds to the terminal Y of FNR.

We subsequently explored mutations of F397 in both full length and DH constructs of SpNOX. The flavin content of purified F397W is much higher than that of WT, while the F397S is not able to retain flavin in its binding site (*Figure 4A*), consistent with our hypothesis about flavin binding. As displayed in *Figure 4B*, in our standard activity conditions, F397W has little or no activity while F397S shows enhanced activity compared to WT. Increasing the NADPH concentration 10-fold resulted in low but significant activity of the F397W mutant (not shown). These results indicate that the W and S mutations of position 397 do not significantly affect the structural integrity of the SpNOX constructs. We used Michaelis-Menten experiments to test the affinity for both NADPH and NADH of F397S mutants of full-length enzyme and DH constructs; such experiments were not possible with F397W because of its very low activity (*Figure 4*). F397S mutants show similar results to WT (*Table 3*) in both constructs indicating that S in this position does not greatly impact affinity for either substrate; this mutation produced an increase in $k_{cat}$ for both constructs, consistent with the hypothesis that S in this position provides increased access of substrate to the catalytic site.

## High resolution structure of the SpNOX dehydrogenase domain

The use of the full-length SpNOX F397W construct resulted in about 20% of the crystals produced displaying diffraction limits close to 4 Å, with some crystals showing an anisotropic diffraction between 3 and 3.3 Å in a privileged direction of the crystals, as often observed for membrane proteins (*Martin et al., 2023*; *Robert et al., 2017*). Despite the improvement of the crystal quality, the anisotropy precluded resolution without a good model for molecular replacement. However, the soluble SpNOX$_{DH}$ domain represents more than 50% of the full-length enzyme, making it a suitable model for use in molecular replacement. We obtained some crystals of both SpNOX$_{DH}$ WT and F397W, co-crystallized with FAD (*Figure 5—figure supplement 1*). Unfortunately, crystals of the SpNOX F397S mutant with or without NADPH could not be obtained.

De novo resolution at 1.94 Å of SpNOX$_{DH}$ F397W was possible because we co-crystallized in the presence of sodium bromide and collected a bromide-SAD dataset (see methods). To solve the SpNOX$_{DH}$ WT structure, we used the SpNOX$_{DH}$ F397W structure as a model, and achieved 2.5 Å resolution. As expected, SpNOX$_{DH}$ presents a typical fold of the FNR superfamily of reductase domain

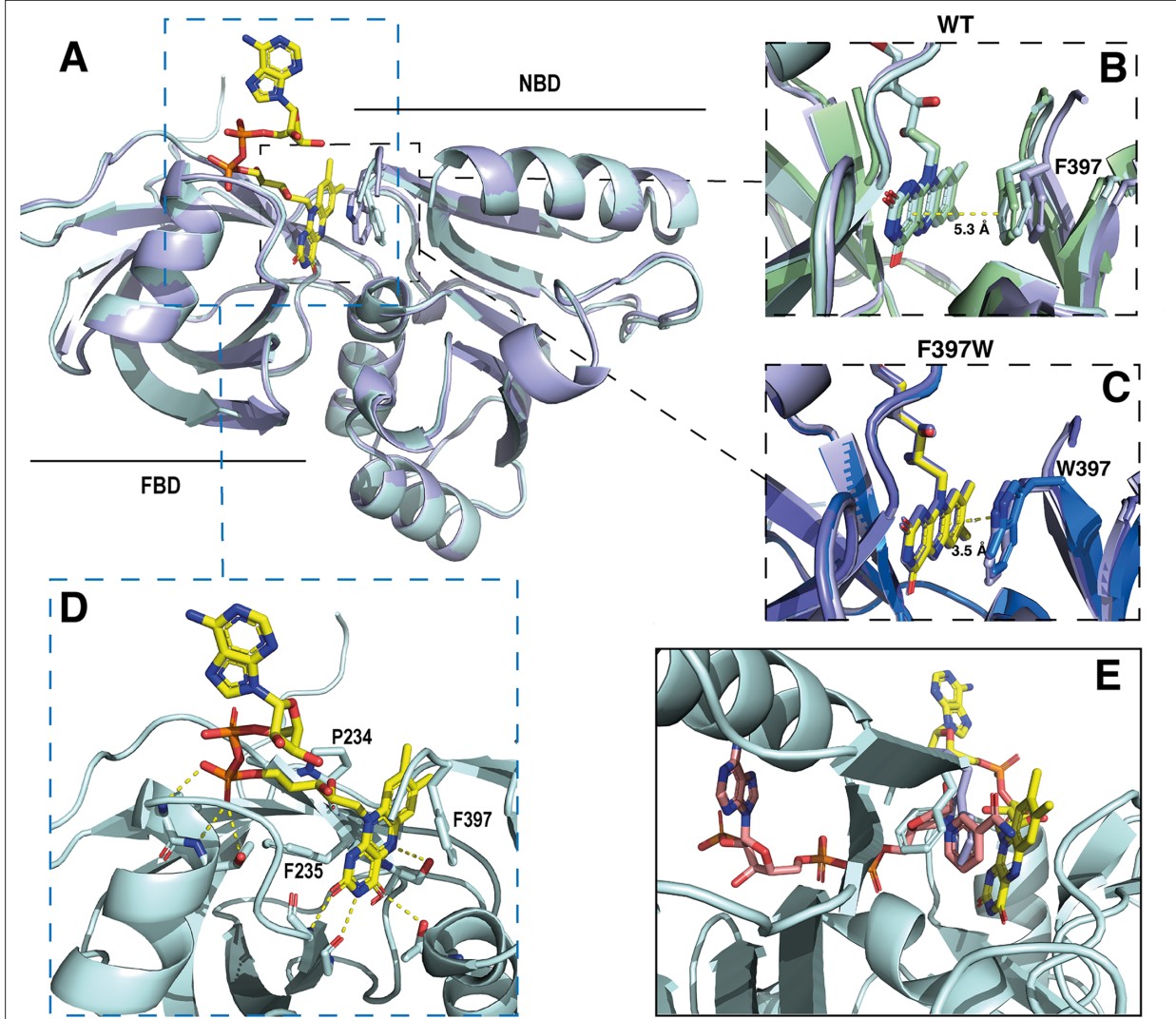

**Figure 5.** Crystal structures of the DH domain of SpNOX WT and F397W. (**A**) Superposition of SpNOX$_{DH}$ WT (pale cyan ribbon, PDB: 8qq5) and SpNOX$_{DH}$ F397W (light blue ribbon, PDB: 8qq1); FAD is shown as sticks colored by atom, side chains of residues in position 397 (respectively F and W), are shown as sticks in the same color as the corresponding ribbon. (**B** and **C**) Superposition of the three different molecules of the asymmetric unit of WT SpNOX$_{DH}$ (**B**) and of SpNOX$_{DH}$ F397W (**C**), zoomed on the interaction of the isoalloxazine ring with the aromatic residue at position 397. (**D**) Close up of the FAD binding site in SpNOX$_{DH}$ WT; polar contacts are shown with dotted lines. Colors as for A. (**E**) SpNOX$_{DH}$ with the F397 (pale cyan sticks) and W397 (light blue sticks) residues superimposed; FAD as in A; and NADPH (salmon sticks) shown based on a superposition of the pea FNR:NADPH complex (PDB: 1qfz) with SpNOX$_{DH}$.

The online version of this article includes the following figure supplement(s) for figure 5:

**Figure supplement 1.** Crystals of SpNOX$_{DH}$ F397W (left) and SpNOX F397W (right).

**Figure supplement 2.** Asymmetric unit of crystal of SpNOX$_{DH}$ WT (top-grey) and of SpNOX$_{DH}$ F397W (bottom-blue).

containing two sub-domains, the FAD-binding domain (FBD) and an NADPH-binding domain (NBD) (*Figure 5A*). Both WT and F397W SpNOX$_{DH}$ display 3 molecules per asymmetric unit and the two structures are identical. In the asymmetric unit of SpNOX$_{DH}$ WT, one of the three molecules contains no flavin (*Figure 5—figure supplement 2*.). In the asymmetric unit of SpNOX$_{DH}$ F397W, all molecules of the unit contain a well-defined FAD (*Figure 5—figure supplement 2*). These observations agree with flavin contents observed in solution (*Figure 4A*) and support the hypothesis that the F397W mutant stabilizes FAD binding.

In strong correlation with the affinity data obtained for the flavins shown in *Table 2*, the DH domain mainly interacts with the isoalloxazine ring through polar contacts with the protein backbone and a

few residues (P234, F235, and S23) belonging to the conserved FAD-1 binding motif (*Massari et al., 2022*; see *Figure 1*). The DH domain shows almost no contact with the ribitol and adenosine parts of the FAD. With the exception of possible polar contact of the first phosphate group with the backbone of the protein, the rest of the FAD molecule points out into the solvent, in the direction of the missing TM domain (*Figure 5D*).

In the wild type, the isoalloxazine ring is sandwiched between FBD residues P234 and F235 on its *si*-face, and the NBD residue F397 on its *re*-face (*Figure 5D*). Superposition of the different SpNOX$_{DH}$ WT molecules from the crystal asymmetric unit shows that the F397 aromatic ring imperfectly occupies the position in front of the *re*-face of the isoalloxazine ring (*Figure 5B*), with orientation 40° with respect to the plane of the isoalloxazine ring, and distance at 5.3 Å. In contrast (and as predicted from the homologous FNR structure), the tryptophan of the F397W mutation makes an extensive and much tighter parallel stacking interaction at a distance of 3.5 Å from the isoalloxazine ring (*Figure 5C*). From these observations, WT SpNOX$_{DH}$ could well behave as a flavin reductase, able to use different flavins as substrate through the sole recognition of the isoalloxazine ring. As *Figure 4A* shows, the F397W mutant may fix the flavin as a cofactor. *Figure 5E* shows the sidechains of both the F and W residue overlaid, on which is superimposed a putative binding mode of NADPH based on the pea FNR structure (*Deng et al., 1999*). Trp at position 397 appears to preclude access of the nicotinamide to the isoalloxazine, preventing direct hydride transfer. Phe in this position, however, leaves more space in this pocket and the sidechain has more degrees of freedom, making it feasible for the nicotinamide to displace it and transfer hydride to the isoalloxazine. These observations most likely explain the drastic difference of activity between WT and F397W; they probably also explain the enhanced activity observed in the F397S mutant which provides even more space near the isoalloxazine (*Table 3*). The increase of the global $k_{cat}$ in the F397S mutant is an additional argument to support hydride transfer as the limiting step in NOX electron transfer.

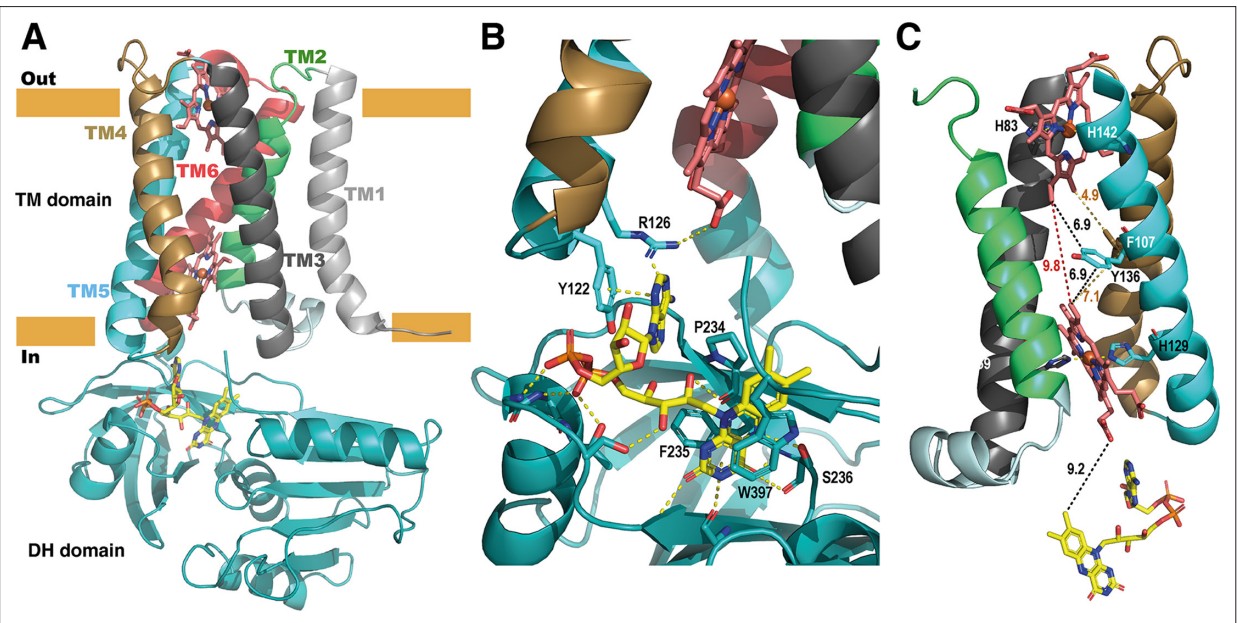

**Figure 6.** Structure of the full-length SpNOX. (**A**) Overall structure of SpNOX F397W at 3.6 Å with ribbons in the TM colored by helix, hemes (red sticks) and FAD cofactor (colored by atom). (**B**) The FAD binding site; FAD as in A, side chains (sticks) of residues involved are labeled and interactions are indicated by yellow dotted lines. (**C**) Electron pathway from FAD to distal heme; for clarity, the DH domain and TM1 and TM6 are omitted. Two alternative electron pathways are indicated, from FAD to heme (both depicted as in A), through a direct transfer between the hemes (red dotted line) or with a relay between the hemes using either the well conserved aromatic residue Y136, or F107, both at intermediate distance from the hemes (black and orange dotted lines); distances in Å are indicated.

The online version of this article includes the following figure supplement(s) for figure 6:

**Figure supplement 1.** Packing in crystal of SpNOX.

## Structure of full-length SpNOX

The high-resolution model of the DH domain of SpNOX allowed resolution of the anisotropic crystals of full-length SpNOX-F397W to 3.6 Å. Small extracellular loops limiting crystal contacts together with intrinsic interdomain flexibility of SpNOX are probably the source of the crystal anisotropy due to subtle variation of packing within the crystals (*Figure 6—figure supplement 1*). Like the previously reported structures of the NOX family (*Liu et al., 2022*; *Magnani et al., 2017*; *Sun, 2020*; *Wu et al., 2021*), the crystallographic model of SpNOX harbors the N-terminal TM domain that encompasses 6 membrane-spanning alpha helices chelating two hemes, and the C-terminal cytosolic DH domain (*Figure 6A*). The density corresponding to the two heme groups is packed in the channel formed by TM helices 2–5. The inner and outer heme are respectively coordinated by the imidazole rings of pairs of histidines H69 (TM3)/H129 (TM5) and H82 (TM3)/H142 (TM5), holding them perpendicular to the plane of the membrane (*Vermot et al., 2020*). The metal-to-metal distance of the two hemes is ~21.5 Å, while the shortest interatomic distance is ~9.8 Å, between the vinyl groups.

As observed in previously solved structures of NOX enzymes (*Liu et al., 2022*; *Magnani et al., 2017*; *Sun, 2020*; *Wu et al., 2021*), a similar electron transfer relay across the membrane was mapped. SpNOX has 9.2 Å edge-to-edge distance between the isoalloxazine ring to the inner heme; a conserved aromatic residue, Y136, at equal distance from both hemes (6.9 Å), or even F107 also on the pathway to the hemes, could assist the electron transfer between the hemes as previously suggested (*Magnani et al., 2017*).

## FAD binding site in the full-length SpNOX: role of the D-Loop region

We observed electronic density corresponding to the FAD cofactor at the interface of the DH and TM domain. FAD binding in the full-length construct preserves the π-stacking interaction with W397 and the interactions with F235 and P234 observed in the DH-only structure. In addition, the full-length SpNOX structure shows that the adenine extremity of FAD is packed in a pocket including the amino acid F399 from the DH domain– providing a potential role for this residue – as well as residues from the D-loop of the TM domain. Indeed, Y122 and R126 could stabilize the adenine ring simultaneously. Y122 interacts through π-stacking with the adenine ring (*Figure 6B*). R126 is at a reasonable distance to bridge with the adenine ring and simultaneously a carboxylate of the proximal heme. However, the resolution of the structure makes the accurate positioning of side chains difficult, particularly in this region which displays large electron densities.

To check the validity of our side-chain observations we generated alanine mutants of Y122 and R126 and checked the affinity of several flavins for each mutant (*Figure 6B* and *Table 4*). We determined affinity for NADPH for all mutants; similar affinity in all mutants suggests no effect of these mutations on global enzyme structure (*Table 4*). The results in *Table 4* shows that only the affinity for FAD is impacted by Y122A and R126A, confirming the role of these residues in the interaction with the FAD's adenine ring, which is missing in the other flavins tested. Y122A diminishes FAD affinity by a factor of 6 and R126A by a factor of 23. This latter suggests that R126 is crucial for FAD affinity;

**Table 4.** Determination of $K_m$ for flavins as a function of the mutations in the D-loop region.
Data here were obtained in the optimal buffer for SpNOX rather than the co-optimized buffer for comparing full-length and DH constructs. It has no impact on flavin $K_m$ determination (compare WT line with values from *Table 2*) but $K_m$ of full-length SpNOX for NAD(P)H are lower by a factor 2–3 compared to the co-optimized buffer. For the corresponding curve see *Table 4—source data 1 and 2*. The sample sizes for the activity measurements were between 2 and 3.

| | FAD | | FMN | | Riboflavin | | NADPH | |
|---|---|---|---|---|---|---|---|---|
| | $K_m$ | $\frac{K_m^{mut}}{K_m^{WT}}$ | $K_m$ | $\frac{K_m^{mut}}{K_m^{WT}}$ | $K_m$ | $\frac{K_m^{mut}}{K_m^{WT}}$ | $K_m$ | $\frac{K_m^{mut}}{K_m^{WT}}$ |
| WT | 0.06±0.01 | - | 0.9±0,3 | - | 6.4±0.2 | - | 13±0.74 | - |
| Y122A | 0.38±0.5 | 6.4 | 1.5±0.9 | 1.7 | 9.6±0.08 | 1.5 | 23.8±0.04 | 1.8 |
| R126A | 3.5±0.33 | 59,1 | 1.5±0.9 | 3.2 | 6.63±0.38 | 1.03 | 24.4±0.45 | 1.87 |

The online version of this article includes the following source data for table 4:

**Source data 1.** Michaelis Menten analysis of SpNOX and SpNOX Y122A and SpNOX R126A as a function of the flavin substrate.

**Source data 2.** $k_{cat}$ determined as a function of the mutation in the D-loop and the flavin used.

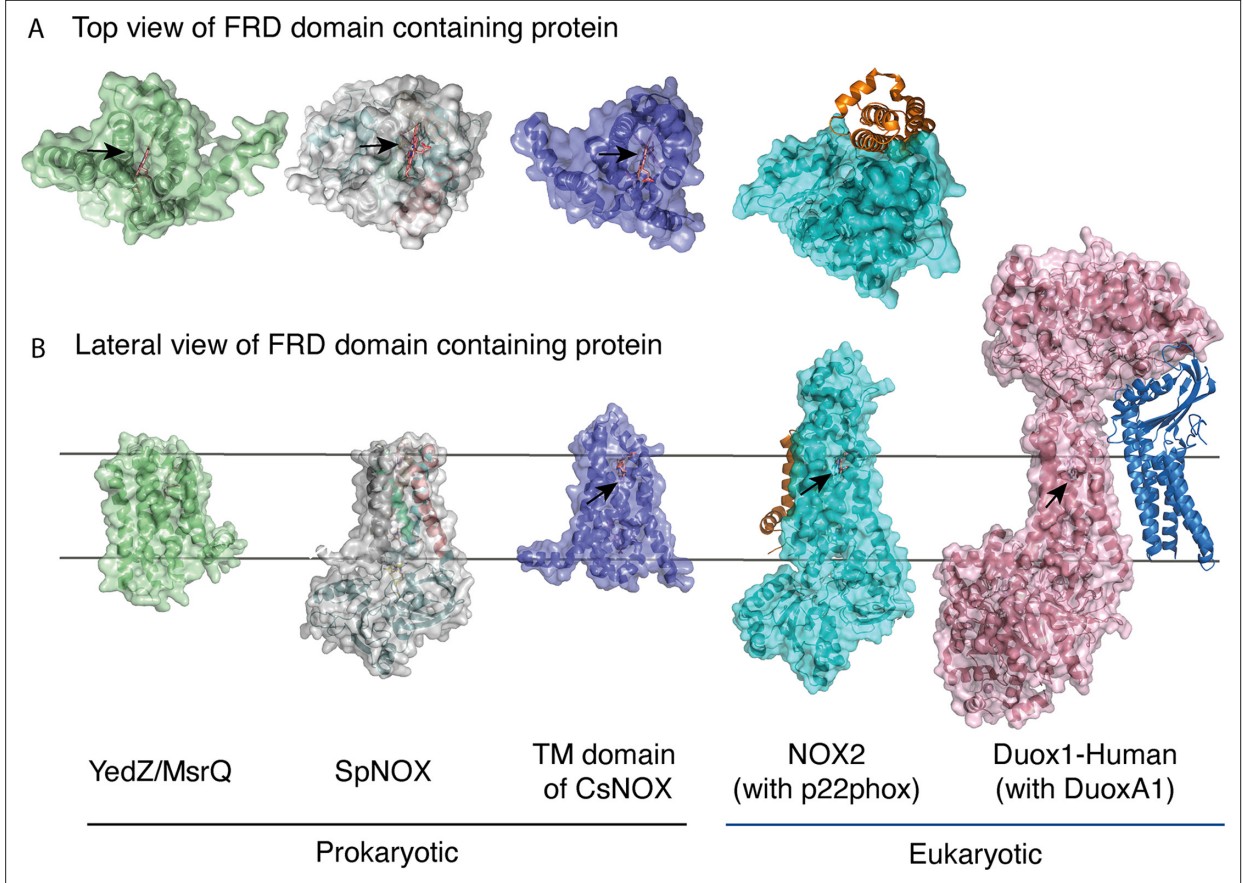

**Figure 7.** FRD containing proteins, represented as surfaces, and accessory proteins (p22phox and DUOXA1) represented as ribbons. TM2 to TM5 were structurally aligned; all proteins are depicted in the same orientation based on this structural alignment. (**A**) Top (extracellular) views of the TM domains of MsrQ (AlphaFold model: P76343, green), SpNOX (this work, PDB: 8qq7, gray), TM domain of CsNOX (PDB: 5O0t, purple), NOX2/p22phox (PDB: 8gz3, blue) and DUOX1 (PDB: 7d3f, pink). (**B**) Each protein's lateral side view (a 90 °C rotation from the orientation in A). Black arrow shows potential access to distal heme either from top or from lateral side. Hemes are represented as salmon sticks.

possibly the simultaneous interaction between R125 and both heme and flavin (proposed above) could hold the adenine ring in a position to be sandwiched between Y122 on one side and F399 on the other. Surprisingly, while the position corresponding to Y122 is very highly conserved as an aromatic residue in the full NOX enzyme family, the position corresponding to SpNOX R126 shows a difference between eukaryotes and prokaryotes. In eukaryotes, the position corresponding to R126 is very highly conserved as an aromatic, while in prokaryotes it is most often a basic residue, although aromatics are not uncommon. The interactions with the TM domain's D-loop (or lack thereof) provide an explanation for the differences among the affinities of the flavins for the full-length protein (*Table 2*).

## Access to the distal heme: diverse situations from prokaryotic to eukaryotic enzymes

The FRD found in NOXs but also in MsrQ and STEAP proteins, transfers electrons from the cytoplasmic to the extracellular side of the cell. Thus, access of the distal heme is essential to the specificity of the electron acceptor and its product. *Figure 7* compares the distal heme access of a panel of FRD-containing proteins from prokaryotes and eukaryotes. From the structures available to date, including the present study, some FRDs (MsrQ, SpNOX, and CsNOX) exhibit small extracellular loops and free access to the distal heme (*Figure 7A*), while others (NOX2) have long extracellular loops (*Figure 7A*) and sometimes another extracellular domain (DUOX) (*Figure 7B*) create a cap that limits access. Homology models of both NOX4 (*O'Neill et al., 2018*) and NOX1 (*Ward et al., 2023*) also predict similar caps. In NOX with large caps, the distal hemes are deeply masked, implying that $O_2$ must diffuse into the protein. On the basis of structures, several potential tunnels have been proposed

for NOX2 and DUOXs (*Liu et al., 2022*; *Noreng et al., 2022*; *Sun, 2020*; *Wu et al., 2021*) and for the homology model of NOX1 (*Ward et al., 2023*). For most of the tunnels identified, bottleneck constriction would require protein dynamics to allow diffusion and the exact pathway for $O_2$ still remains to be defined. However, a potential entry site with direct access to the distal heme from the lateral side can be seen in the NOXs with large caps (*Figure 7B*). In addition, an internal pocket in close proximity to the distal heme has been proposed as the site of $O_2$ reduction through an outer sphere electron transfer mechanism. This pocket is characterized in eukaryotic enzymes by a conserved His and Arg involved in $O_2$ binding and heme interaction; the residues characteristic of the $O_2$ reducing center in eukaryotic FRD domains of NOX and DUOX enzymes are not conserved in SpNOX. The importance of the conserved His and Arg in the reducing center pocket of eukaryotes are strongly supported by CGD-induced mutations (*Cross et al., 1995*; *Picciocchi et al., 2011*; *Rae et al., 1998*). A pathological NOX1 variant was used to show that an additional Asn residue, separated from the conserved His by two residues, is necessary for efficient $O_2$ reduction *Ward et al., 2023*; this Asn is highly conserved in animal NOX (see Helix 3 motif in *Figure 1*) but not prokaryotes. Entry site, tunnel size and reducing center pocket location ensure the selectivity to $O_2$ as unique electron acceptor for NOXs with large caps.

FRD containing proteins with direct access to the distal heme may reflect more versatility with regard to final electron acceptors. MsrQ, to date the only FRD-containing prokaryotic protein with clearly identified physiological function, provides an interesting case. In *E. coli*, MsrQ transfers its electrons to MsrP, a periplasmic methionine sulfoxide reductase, which repairs proteins damaged by methionine oxidation (*Gennaris et al., 2015*). The absence of long extracellular loops allows MsrP to dock on MsrQ and to receive electrons directly from the distal heme to its molybdopterin center. Similarly, SpNOX's exposed heme may confer its ability to directly reduce Cyt. *c* (*Hajjar et al., 2017*). The physiological functions of neither SpNOX nor CsNOX are known; in addition to their NOX activity, they may use electron acceptors other than $O_2$.

## Domain orientation in SpNOX structure: comparison with other NOX structures

The previous crystal structure of a prokaryotic NOX was in fact two independent structures of the separate TM and DH domains of CsNOX (*Magnani et al., 2017*). With the SpNOX structure presented here we provide the first entire prokaryotic NOX structure, affording a view of the orientation of the TM domain relative to the DH. The importance of the relative orientation of the domains in NOX proteins has been underlined as a critical point in the activation and the efficiency of electron transfer (*Liu et al., 2022*; *Wu et al., 2021*). Comparison of the recent structures of an inactive 'resting state' NOX2 and of the presumably active 'high $Ca^{2+}$ state' DUOX1, emphasized the large 49° rotation of the DH domain of one structure relative to the other. This motion leads to a 12.4 Å movement of the FAD, which was suggested as critical for the activation of the electron transfer (*Liu et al., 2022*). Including SpNOX into this movement analysis is all the more interesting given that SpNOX is constitutively active. Here, *Figure 8A* shows that in our SpNOX structure, the relative position of TM *vs* DH domain is intermediate between the inactive (NOX2) and active (DUOX) structures. *Figure 8B* details the potential intermediate motions of SpNOX: the relative position of SpNOX DH with respect to the inactive state NOX2 DH rotates 13° and translates 5 Å toward the plane of membrane, while the relative position of the high $Ca^{2+}$ 'activated' state of DUOX1 DH rotates almost an additional 49° and translates an additional 7.6 Å. SpNOX's domain orientation is much closer to that observed in the NOX2 resting state than that of DUOX1.

The relative positions of the cofactors involved in the first steps of electron transfer are highlighted in *Figure 8C*. The distance between hemes in a given FRD domain is invariant; thus, the efficiency of electron transfer depends on two main factors: the distance between the nicotinamide of NAD(P)H and the isoalloxazine ring of FAD, and the distance between the isoalloxazine ring of FAD and the proximal heme. Surprisingly, there is no evident correlation between DH motion from inactive to active state and contraction of the distance between FAD and heme. In fact, this distance actually increases between inactive NOX2 and constitutively active SpNOX, and from SpNOX to 'activated' (high $Ca^{2+}$) DUOX. Nonetheless, in all cases these distances, ranging from 7.2 to 10.2 from edge to edge (from methyl group of the flavin ring to the carboxylate of the heme), all remain compatible with biologically relevant electron tunneling probability, calculated as $\leq 14$ Å. (*Page et al., 1999*).

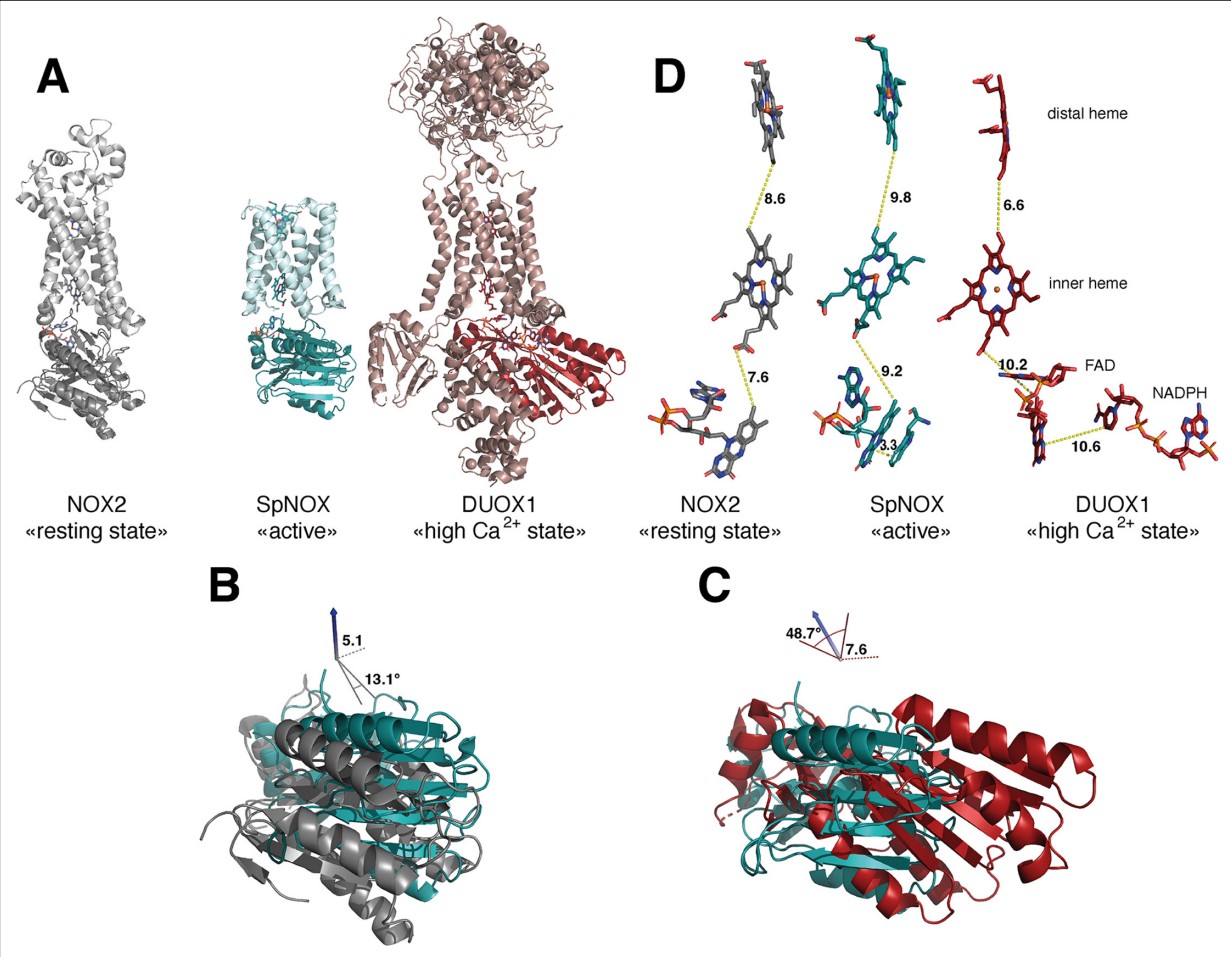

**Figure 8.** Comparison of domain and cofactor organization in NOX2, SpNOX and hDUOX1. (**A**) Proteins were superimposed and are shown in identical orientations based on superposition, resting state NOX2 (PDB: 8gz3, gray ribbon), active SpNOX (this work, PDB: 8qq7, green ribbon) and high Ca$^{2+}$ DUOX1 in (PDB: 7d3f, red ribbon), with DH domains in darker color. (**B** and **C**) DH domains only are represented in same orientations and color as in A. In B, angle and translation from NOX2 to SpNOX$_{DH}$ are indicated. In C, angle and translation from SpNOX$_{DH}$ to DUOX1 are indicated. (**D**) Cofactors necessary for electron transfer are shown as sticks, in identical orientations as in A-C, with distances indicated. In SpNOX the sidechain of Trp397 is shown and the distance to FAD isoalloxazine ring indicated. For DUOX1, NADPH is shown and the distance of nicotinamide to the isoalloxazine indicated.

These observations led us to question the paradigm that switching between resting and activated states requires DH motion. The lack of DH motion between the low and the high Ca$^{2+}$ state human DUOX1 structures adds further hesitation. The DH domains of these two structures superimpose with a RMSD of 0.246 Å, and the FAD and NADPH of these structures also superimpose perfectly. The distance between the nicotinamide ring of the NADPH and the isoalloxazine ring of the FAD even in the activated DUOX1 structure is surprisingly large for efficient hydride transfer (*Figure 8D*). In line with conclusions from DH homologs (*Deng et al., 1999*; *Hermoso et al., 2002*), our data (*Table 3*, *Figures 4 and 8*) suggests that the SpNOX F397W sidechain represents the position that would be occupied by the nicotinamide ring of NADPH. The distance between the Trp397 sidechain and the isoalloxazine ring of FAD in our structure is 3.3 Å, similar to the corresponding distance of these moieties in homologous DH domains and suitable for hydride transfer. In the case of DUOX1, it has been suggested (*Wu et al., 2021*) that to enhance hydride transfer, the 'high-Ca$^{2+}$ activated' state requires 'tensing' of the DH from a relaxed orientation, which can be envisioned as the relative motion of the NBD subdomain (containing NADPH) toward the FBD (containing FAD), rather than a large swing or translation of the entire DH domain relative to the TM domain.

These considerations suggest that differences in relative domain orientations in NOXes might represent isoform specificity rather than intermediate activation states. If so, these differences and

those of the cofactor distances could fine tune the electron transfer properties and resulting efficiency of ROS production, linking the isoform with its physiological context. In this context, the impact of the lipid environment on domain orientation is suggested by longstanding observations of lipid effects on NOX activity (***Koshkin and Pick, 1993***; ***Shpungin et al., 1989***) and by the observation of a lipid in the murine DUOX structure that mediates the interaction between the TM and DH domain through a direct connection with the phospho-ADP-ribose part of the NADPH (***Sun, 2020***). Structures solved in detergent like ours presented here may be released from interaction with the membrane, increasing flexibility. The question of what happens in NOX2 –whether DH tensing, domain motion, or something else – remains elusive and awaits further structural characterization. In any case the SpNOX domain orientation presented here makes it a rather close model of human NOX2.

## A close-up view of NOX's NAD(P)H binding domains *vs* the FNR gold standard

The striking difference in distance between NADPH and FAD at the active site (***Figure 8D***) parallels a structural difference between NOX2 and DUOX1 on one hand, and SpNOX on the other. Here, we use the well-characterized FNR structure as a basis of comparison for DH domains. The NADPH binding groove has characteristic motifs universally conserved in DH domains (***Piubelli et al., 2000***; motifs NADPH1 and NADPH4 in ***Figure 1***); these motifs provide specific contacts with the NADPH ligand, while the critical aromatic in the C-terminal motif is important for FAD stabilization and apparently displaced by NADPH binding (***Deng et al., 1999***; ***Kean et al., 2017***). In FNR, the position of the NADPH1 and NADPH4 motifs – representative of the full binding groove for NADPH – are separated from the flavin binding site by a narrow space (***Figure 9*** top left), while in DUOX1 the corresponding space is much wider (***Figure 9***, top right). Superposition of the four DH domains makes it clear that in SpNOX the distance between the binding grooves for the two ligands is short, like in FNR (***Figure 9*** middle left), while the distance between the ligand grooves in NOX2 is longer, like in DUOX1 (***Figure 9***, middle right). NADPH-binding residues of FNR are mostly conserved in SpNOX (***Figure 9—figure supplement 1***), while the two Arg residues that interact with the 2'-phosphate of NADPH in DUOX 1 are conserved in NOX2, although the side chain of one of those (R446) is not visible in the CryoEM structure of NOX2 (***Figure 9***, bottom left).

Although the SpNOX and NOX2, in the resting state, structures lack NADPH, superposition produces an excellent picture of the location of the NADPH in these proteins. ***Figure 9*** (bottom) shows that when the proteins are superimposed, the NADPH from pea FNR fits extremely well into the NADPH binding groove of SpNOX and reproduces the short distance between nicotinamide and isoalloxazine seen in FNR. Similarly, the superimposed DUOX1 NADPH fits very well into the corresponding binding groove of NOX2. Even this crude docking with no energy minimization leaves little doubt about the overall NADPH position in SpNOX and NOX2. Our functional characterization of mutants in position 397 strongly support the picture presented here. Of course, new structures for NOX2 and SpNOX in complex with NAD(P)H would be a welcome addition to confirm these observations.

Taking into account, the similar interdomain organization and comparable electron pathways (***Figure 8D***) of SpNOX and NOX2, the different topology at the interface of their respective FBD and NBD subdomains would explain the switch from an inactive conformation of NOX2 in its resting state to a constitutively active conformation of SpNOX. This idea is consistent with the idea that electron transfer in the high $Ca^{2+}$-state DUOX1 structure may require a 'tensing' motion that brings the two subdomains, and hence their ligand binding grooves, closer together (***Wu et al., 2021***). The mechanism by which the inactive resting state of NOX2 changes to active, and how this situation fits with the activated state of the DUOX1 structure, remains to be elucidated.

## Conclusion

The important roles of NOX in human, animal, and plant health and disease explain the field's focus on eukaryotic NOX. However, prokaryotic NOX's advantages in production and handling offer opportunities to decipher structural and mechanistic details relevant to the whole family (eg. ***Magnani et al., 2017***; ***Vermot et al., 2020***). We present here extensive enzymological characterization of SpNOX activity and also new structural information on both full-length enzyme and the isolated DH. First, the particular effort to optimize conditions for SpNOX$_{DH}$ allowed the investigation of the first

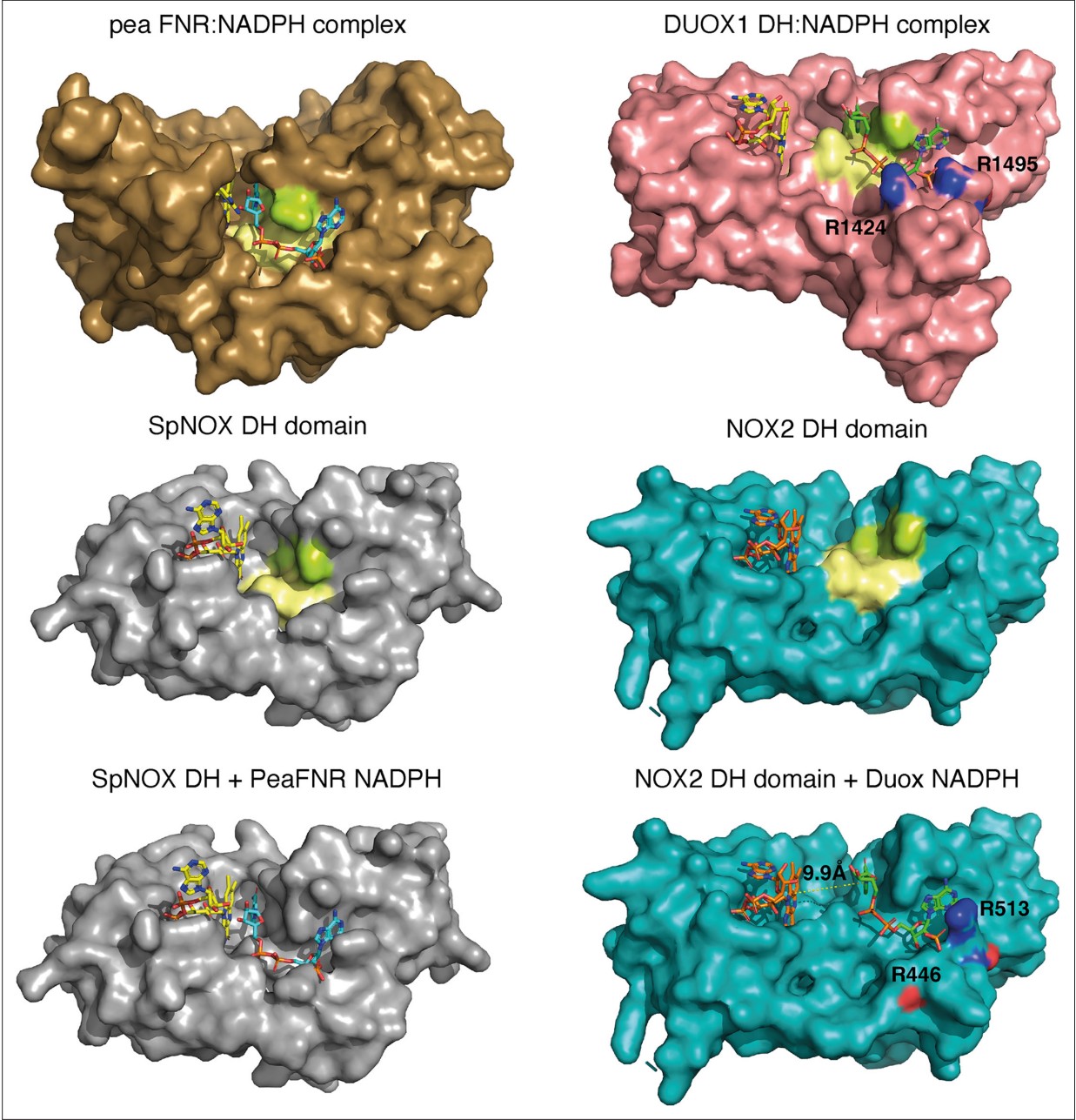

**Figure 9.** Comparison of FNR domains and their putative NADPH binding sites. Top, pea FNR (PDB: 1qfz), DUOX1 domain (PDB: 7d3f) in complex with NADPH. Middle, SpNOX DH domain (PDB: 8qq7) and NOX2 DH domain in the resting state (PDB: 8gz3). Bottom, SpNOX DH is represented with the NADPH from pea FNR after superposition of those proteins; NOX2 DH is represented with the NADPH from DUOX1 after superposition of those proteins. Arg residues conserved between DUOX1 and NOX2 are highlighted using CPK colors, side chain of R446 is not defined in NOX2 CryoEM structure. In top and middle line, pale yellow and lime patches represent NADPH motifs 1 and 4, respectively (see *Figure 1*). Assuming that NADPH binding implies movement of F397, residues [397]FKF[399] have been omitted in the SpNOX DH surface to increase clarity and visibility of NAD(P)H binding groove.

The online version of this article includes the following figure supplement(s) for figure 9:

**Figure supplement 1.** Overlay of the pea FNR (sand) and the DH domain of SpNOX (pale cyan).

**Figure supplement 2.** Superposition of NOX2 in the resting state and SpNOX.

step of NOX electron transport, hydride transfer from NAD(P)H to FAD (*Figure 2* and *Table 1*). We showed that NADPH and NADH are equivalent electron donors for SpNOX, constituting a significant difference from eukaryotic NOX. Comparison with the full length SpNOX demonstrates that NAD(P)H binding is largely independent from the TM domain, at least in our conditions in which SpNOX is solubilized in LMNG detergent. Comparison of $k_{cat}$ obtained from both constructs suggests that the hydride transfer from NAD(P)H to FAD is the rate limiting step in the overall NOX reaction (*Table 3*). Structural and functional characterization of WT and mutant F397W and F397S highlight the critical role of the aromatic at 397: Phe provides flavin isoalloxazine ring interaction and stabilization and also properties that allow entrance of the nicotinamide ring to the active site, similar to the mechanism long accepted in FNR family domains. Parallel analysis of flavin analogs using both DH and full-length SpNOX shows that the DH domain alone has no selectivity for flavin type. SpNOX, on the other hand, displays a marked selectivity for FAD with affinity at the nM level, indicating binding contribution from the TM domain. This is supported by interactions visible in the entire SpNOX structure, and a related mutational analysis, highlighting the role of the D-Loop region in the interaction with the adenine ring of the flavin (*Figure 6B*).

Structural comparison of NOX structures indicates that accessibility of the distal heme of SpNOX and CsNOX may reflect plasticity in the final electron acceptor, while the inaccessibility of other NOXes may enforce $O_2$ as a substrate (*Figure 7*). On the other hand, this comparison also generates a new conundrum, how to explain hydride transfer in activated states of NOX. SpNOX shares similar domain organization with NOX2, and the orientations and distances between cofactors in the electron transfer pathway are similar (*Figure 9—figure supplement 2* for overlay of both structures). Surprisingly, DUOX1 domain organization and cofactor distances are quite different, raising the possibility that domain organization is isoform specific. Considering the constitutive activity of SpNOX, we believe that the electron pathway as organized in NOX2 is poised for electron transfer, and that the difference in position of the NADPH binding groove could explain the difference between active and inactive states. The 'high $Ca^{2+}$' activated state of DUOX1 also displays the distant binding groove suggesting that an additional step might be required to reach an efficient active state. Rather than a global repositioning of the DH, activation of NOX2 may require an internal rearrangement within the DH that brings the NADPH groove closer to the FAD site; a similar rearrangement may be necessary for electron transfer in activated DUOX1. Conformational change within the DH may agree with a previous study concluding that p67[phox], which activates NOX2, regulates hydride transfer from NADPH to the FAD rather than NADPH binding. (*Nisimoto et al., 1999*). In this view, the assembly of p67[phox] with the DH could promote the required rearrangement within the DH domain. However, the resting state of NOX2 has been reported to resist initial binding of NADPH (*Liu et al., 2022*). This could be a second level of control for NOX2.

During the review process of this work, another group published a pre-print presenting several CryoEM structures of SpNOX (*Dubach et al., 2023*). Remarkably, considering the small size of the protein and its solubilization in detergent, they solved several structures from 2.33 to 2.75 Å resolution. Their high-resolution in the TM domain allowed them to improve the orientation of the adenine part of FAD and of the distal heme. However, apart from these adjustments, their Cryo-EM and our crystal structures are essentially the same (residus Y122 of the D loop is stabilizing the adenine from FAD, the distances between the cofactors are the same as well as the access to the active site or the distal heme). The work of *Dubach et al., 2023* includes both functional data and additional structural data that complements ours. For example, they mutated F107, one of the potential aromatic relays for the electron transfer between the hemes. This mutation did not produce any functional impact, supporting the idea of a direct electron transfer from the proximal to the distal heme, although only simultaneous F107 and Y136 mutations would definitively close the interheme 'relay' possibility. An important contribution of their Cryo-EM work includes structures of SpNOX with bound NAD(P)H; these fully validate the binding site we anticipated (*Figure 9—figure supplement 1*), and confirm the critical roles of residue K250, S348 and Y353 in NAD(P)H binding. In addition, these structures highlight the capacity of the active site to accommodate the F397 side chain, the nicotinamide ring and the flavin ring within the groove where hydride transfer should occur. Their structures document the role of F397 in positioning the nicotinamide core within this groove and in controlling hydride transfer, since the F sidechain is located between the flavin and nicotinamide rings. In one of the NADPH-containing structures, they used a F397A mutation, which

mimics the situation in which the F sidechain has moved out from between the nicotinamide and flavin rings, to trap a nicotinamide conformation productive for hydride transfer, approaching 3.6 Å from the isoalloxazine ring. These data are fully consistent with our enzymatic and structural characterization of F397 mutants. Altogether both our structures and theirs strongly support the critical role of F397 in the tuning of the hydride transfer mechanism. Taken together, the observations from both the Cryo-EM and our crystal structures of SpNOX reinforce the need for close proximity of the NADPH and FAD binding grooves for hydride transfer, exemplified in this constitutively active enzyme; this in turn supports our hypothesis, presented herein, that the activated state of *eukaryotic* NOX requires a specific relative motion of the NBD toward the FBD to achieve a productive topology in the DH domain.

In conclusion, SpNOX seems to be a good model of an activated form of NOX2. The molecular tools and structures provided in this work will allow further investigations to understand additional molecular features at work in those fascinating NOX enzymes.

Between the last revision and the proof-reading process of this article, a CryoEM structure of an activated form of neutrophil NADPH oxidase was published in Nature by *Liu et al., 2024*. It illustrates and magnificently lends credence to all the hypotheses, arising from this work on SpNOX, showing a relative movement of the NBD towards the FBD subdomain and thus allowing productive binding of NADPH. This confirms SpNOX, constitutively active, as a model of the activated form of NOX2 and thus gives all the more importance to the mechanistic characterization presented in this article.

# Materials and methods
## Structure-based sequence alignment
A large set of 549 NOX homolog sequences distributed about evenly between prokaryotes and eukaryotes, was aligned using MAFFT (*Katoh et al., 2019*). The sequences were split into transmembrane (TM) and dehydrogenase (DH) domains identified by homology and including 30 overlapping residues between the domains. Each domain was realigned separately with PromalS3D (*Pei et al., 2008*). The TM alignment was constrained to the crystal structure of CsNOX TM domain (5O0T) while the DH alignment was constrained to the crystal structure of CsNOX DH domain (5O0X) and DH domain homologs 1GJR, 1FNB, 1GVH, 2EIX, 3A1F; all other parameters were at default values. The resulting multiple alignments were inspected in Jalview (*Waterhouse et al., 2009*); those with missing secondary structural elements or with unusually large insertions were removed and remaining sequences were pruned to 91% (TM domain) or 95% (DH domain) identity. The final sequence sets (172 sequences of TM and 207 sequences of DH domain) were realigned in Promals3D with default parameters and the same structural constraints specified above. Logos of subsequences of the full alignment were created in WebLogo (*Crooks et al., 2004*).

## Cloning of SpNOX, SpNOX$_{DH}$ WT, F397S and F397W
The synthetic SpNOX gene optimized for expression in *E. coli* including a polyhistidine tag and thrombin cleavage site at its N-terminus was amplified *via* PCR and subcloned into pET-30b (Novagen) between the NdeI/ BamHI sites (*Vermot et al., 2020*).

The expression vector encoding His-tagged SpNOX$_{DH}$ - aa 181–417 of full-length SpNOX - was obtained from SpNOX by site-directed mutagenesis according to the manufacturer's protocol (Quik-Change Lightning Site-Directed Mutagenesis Kit, Agilent), using forward primer 5'gtctggtcccgcgtgg cagtaaaattagctttccgtatctggg3' and reverse primer 5'cccagatacggaaagctaattttactgccacgcgggacca gac3'.

F397S or F397W mutants of SpNOX and SpNOX$_{DH}$ were created using forward primer 5'ACGG AACTGATCTACGAAGGCTCTAAATTCAAATGAGAATTCGAGC3' and reverse primer 5'GCTCGAAT TCTCATTTGAATTTAGAGCCTTCGTAGATCAGTTCCGT3' for F397S or forward primer 5'CGGAACTG ATCTACGAAGGCTGGAAATTCAAATGAGAATTCGAG3' and reverse primer 5'CTCGAATTCTCA TTTGAATTTCCAGCCTTCGTAGATCAGTTCCG3' for F397W.

We retain the nomenclature 'F397x' for these mutations in both SpNOX and SpNOX$_{DH}$ constructs, although in SpNOX$_{DH}$ construct that residue is actually in position 217.

## Production of SpNOX, SpNOX$_{DH}$ WT, F397S and F397W

The expression of SpNOX WT, F397S and F397W was performed as described in *Vermot et al., 2020* for the full-length WT SpNOX.

The DH-only domain proteins SpNOX$_{DH}$ WT, -F397S and -F397W were over-expressed in *E. coli* BL21(DE3) using TB medium supplemented with 50 µg.ml$^{-1}$ kanamycine. At culture OD$_{600}$=1, expression was induced by the addition of 0.5 mM IPTG and the culture was grown overnight at 18 °C. Cells were harvested by centrifugation (5000 × $g$, 20 min, 4 °C) and stored at –20 °C. The purification procedure described below is the final optimized version for SpNOX$_{DH}$ stabilization. The pellet was resuspended in lysis buffer (50 mM TRIS pH7, 1 M NaCl, 10 mM Imidazole and 5% glycerol) supplemented with deoxyribonuclease and a mix of protease inhibitors (chymostatin, leupeptin, antipain, pepstatin at 1 µg.ml$^{-1}$, aprotinin at 5 µg/ml), disrupted by sonication, and cleared by centrifugation at 4 °C (18,500 rpm, JA-25.50 Beckman, 20 min,). The clarified extract was purified on a 4 ml His60 Ni-IDA Superflow resin (Ozyme) previously equilibrated in high salt buffer (HS buffer; 50 mM TRIS pH7, 1 M NaCl, 10 mM Imidazole and 5% glycerol). After an extensive wash (20 column volumes, HS buffer), the protein was eluted with 50 mM TRIS pH7, 1 M NaCl, 300 mM Imidazole and 5% glycerol and the buffer was exchanged against a stabilization buffer (50 mM Bis- TRIS propane pH6.5, 0.3 M NaCl, 5% glycerol) *via* gravity using a desalting column (PD10 17-0851-01, GE Healthcare). The selected fractions were pooled and loaded onto a Superdex75 size exclusion chromatography (GE Healthcare) equilibrated in the same buffer. The protein was eluted as a monomer and concentrated by ultrafiltration (10 kDa centrifugal filter unit, Amicon). The protein purity was checked on SDS-PAGE and the concentration was determined by UV-visible spectroscopy using respectively ε$_{280nm}$ = 24870 M$^{-1}$.cm$^{-1}$ for SpNOX$_{DH}$ WT and SpNOX$_{DH}$ F397S or ε$_{280nm}$ = 30370 M$^{-1}$.cm$^{-1}$ for SpNOX$_{DH}$ F397W respectively.

## Thermostability assays for SpNOX$_{DH}$ stability improvement

Thermal stability assays were performed using a Prometheus NT.48 nanoDSF instrument (NanoTemper Inc, Germany). Fifty µl of sample per condition were prepared to perform experiments in triplicates using standard-treated glass capillaries. The excitation at 280 nm was optimized to yield emission intensities of intrinsic fluorescence at 330 and 350 nm, in the absence of heat gradient, in the range of 5000–16,000 A.U. The temperature gradient was set to an increase of 1 °C/min in a range from 20 °C to 95 °C. Protein unfolding was measured by detecting the temperature-dependent change in the ratio of fluorescence emission at 350 nm vs 330 nm. Samples used for testing pH impact on stability were done by diluting 10 times concentrated SpNOX$_{DH}$ solution (2 mg.ml$^{-1}$) to a 0.2 mg.ml$^{-1}$ protein solution in the following 0.1 M buffers depending on the tested pH: citric acid buffer for pH 4, 5, 5.5 and 6; sodium acetate for pH 4.5; bis TRIS-propane for pH 6.5 and 8, Pipes for pH 7; TRIS for pH 7 and 7.5; Bicine buffer for pH 9 and CAPS buffer for pH 9.5.

## Activity assay on purified SpNOX and SpNOX$_{DH}$ domain

The activity of the purified DH domain was measured under aerobic conditions following simultaneously the oxidation of NAD(P)H (decrease of absorbance at 340 nm, ε=6.22 M$^{-1}$.cm$^{-1}$) and the reduction of cytochrome $c$ (increase of absorbance at 550 nm, ε=21.1 M$^{-1}$.cm$^{-1}$), as a function of time, using a Cary 50 UV-visible spectrophotometer (Varian).

A typical assay was as follows: 50 mM bis TRIS-propane pH 6.5, 0.003% LMNG supplemented with 200 µM NADPH, 300 mM NaCl, and 5% glycerol, was added to a 2 mm quartz cuvette. Then 0.5 µg. ml$^{-1}$ SpNOX$_{DH}$ (final) and 10 µM FAD (final) were added, mixing after each addition. For Cyt $c$ reductase assay, 100 µM cytochrome $c$ was added prior to the addition of the protein. Final assay volume was 500 µl. When full length SpNOX was used alone, not in parallel study with SpNOX$_{DH}$, we used a buffer optimized for activity of the full length enzyme: Tris pH 7, 300 mM NaCl, and 0.003% LMNG.

The inhibition of superoxide production was verified by addition of DPI at a final concentration of 50 µM or Superoxide dismutase (10 U) to the reaction mixture after a sufficient time to allow the slope to be characterized in the presence or absence of the inhibitor.

## Data analysis of thermostability and activity assays

The results from thermostability and activity assays were analyzed simultaneously using the JMP statistical software (Version *13.0*. SAS Institute Inc, Cary, NC, 1989–2019) to plot 2D contour plots of Tm and activity as a function of salt concentrations and glycerol. The median of triplicates was used to

compute the graphs, and linear interpolations were done between measurement points to fill the graphs. A 2D linear interpolation was performed on thermostability and activity results as a function of pH and salt concentration using Python to produce a 3D figure.

## Michaelis-Menten kinetic analysis for SpNOX and SpNOX$_{DH}$

All chemicals and reagents were purchased from Sigma-Aldrich. Various NAD(P)H concentrations were obtained by serial dilutions from the saturating NAD(P)H solution in triplicate in a 96-well (flat bottom) Greiner microplate (equilibrated at 25 °C). The reaction was initiated (at 25 °C) by injection of a mixture of the stabilization buffer (50 mM bis TRIS-propane pH6.5, 300 mM NaCl, 10 µM FAD, 5% glycerol and 0.025 mM LMNG) containing SpNox$_{DH}$ (250 ng) and cytochrome $c$ (100 µM, final concentration) or (50 mM Tris pH 7, 300 mM NaCl, 10 µM FAD and 0.025 mM LMNG) containing SpNOX (250 ng) and cytochrome $c$ (100 µM, final concentration) and the absorbance at 550 nm was recorded within 3 min using a monochromator multimode microplate reader (Clariostar, BMG Labtech) after orbital shaking. Michaelis-Menten saturation curves of SpNOX, initial apparent velocities as a function of NAD(P)H concentration, $K_m$ (µM), and $k_{cat}$ (s$^{-1}$) were determined with Prism software. We determined the kinetic parameters for all flavin analogs using a similar procedure, holding NAD(P)H concentration constant at 200 µM and varying the flavin concentrations; similarly, kinetic parameters of SpNOX$_{DH}$ were determined for NAD(P)H based on NAD(P)H oxidation by monitoring the absorbance at 340 nm within 3 min. Affinity is formally described by $K_d$, but in the text we use $K_m$ as a 'relevant approximation of affinity' (*Srinivasan, 2022*).

## Crystallization of SpNOX$_{DH}$ domains

Crystallization experiments were carried out at the High Throughput Crystallization Laboratory (HTX Lab) at EMBL Grenoble using automated protocols (*Cornaciu et al., 2021*; *Dimasi et al., 2007*; *Dupeux et al., 2011*; *Mariaule et al., 2014*). SpNOX$_{DH}$ and SpNOX$_{DH}$ F397W were concentrated to 15 mg.ml$^{-1}$. The crystallization experiments were carried out using the sitting-drop vapor-diffusion method with a crystallization robot (Mosquito, SPTLabtech). A total of 0.1 µL of protein solution and 0.1 µL of reservoir were mixed to equilibrate against 45 µL reservoir solution at 20 °C in 96-well CrystalDirect plates (MiTeGen) and automatically imaged in RockImager robot (Formulatrix). Initially, commercially available screening kits from NeXtal (Classic-suite, PEGs, JCSG) and Molecular Dimension (MemTrans, MemMeso, MemGold, PACT, Morpheus) were used to identify initial crystallization conditions that were optimized in subsequent experiments. Optimal crystals of SpNOX$_{DH}$ grew in crystallization drops with precipitant conditions of 35% PEG-MME500, 0.1 M Sodium Citrate pH5. Optimal crystals of SpNOX$_{DH}$F397W grew in crystallization drops with precipitant condition composed of 17% to 20% w/v PEG3350, 0.1 M Bis-TRIS propane pH 6.5 and 0.2 M Sodium bromide. Crystals of SpNOX$_{DH}$ WT and SpNOX$_{DH}$ F397W typically appeared after 24 hr and reached full size in 7 days with dimensions of respectively 100×20 × 10 µm³ and 80×30 × 20 µm³. Automated high-throughput crystal cryo-cooling and harvesting were performed with CrystalDirect Technology (*Cipriani et al., 2012*; *Márquez and Cipriani, 2014*; *Zander et al., 2016*).

## Data collection and structural solution of SpNOX$_{DH}$ F397W

A bromide-SAD dataset was collected at the macromolecular crystallography beamline ID30B of the European Synchrotron radiation facility, equipped with Pilatus3 6 M detector (*McCarthy et al., 2018*). The wavelength was set above the Br K-edge at 0.9184 Å in order to maximize the anomalous signal, and a highly redundant data set at a resolution of 1.94 Å, including 7200 images with 0.05 degrees oscillation, was collected.

Data processing was performed by the GrenADES parallelproc pipeline (*Monaco et al., 2013*) that is based on XDS (*Kabsch, 2010*) and scaled using aimless *Evans and Murshudov, 2013* from the CCP4-suite (*Winn et al., 2011*). Attempts to solve the structure with the molecular replacement method using homologous models failed. Therefore, we experimentally solved the structure by the SAD method, using the bromide anomalous dispersion signal from crystals grown in presence of 0.2 M sodium bromide. Experimental phasing and initial model building was performed using CRANK2 (*Skubák and Pannu, 2013*) The final model was obtained by successive rounds of manual and automated refinement using the programs REFMAC (*Murshudov et al., 1997*), and BUSTER

**Table 5.** X-ray data collection and refinement statistics of SpNOX$_{DH}$ WT, SpNOX$_{DH}$ F397W and full-length SpNOX F397W.

**Data collection**

| | DH F397W | DH WT | FL F397W |
|---|---|---|---|
| Space group | P4$_1$2$_1$2 | P4$_1$2$_1$2 | P6$_4$22 |
| Cell *a, b, c* (Å) | 104.62, 104.62, 142.68 | 104.88, 104.88, 139.29 | 145.97, 145.97, 153.62 |
| Angles a, b, g (°) | 90, 90, 90 | 90, 90, 90 | 90, 90, 120 |
| Resolution (Å) | 58.94–1.94 (1.97–1.94) | 46.9–2.50 (2.60–2.50) | 47.82–3.62 (3.95–3.62) |
| $R_{merge}$ | 0.117 (1.829) | 0.140 (2.441) | 0.072 (2.1) |
| $I/\sigma$ | 21.2 (2.1) | 18.99 (1.46) | 19.6 (1.7) |
| Completeness (%) | 98.4 (95.6) | 100.0 (100.0) | 47.9 (10.7) 90.9* (85.4*) |
| Redundancy | 13.8 (12.9) | 25.96 (26.86) | 20.4 (20.1) |
| Ellipsoid | n/a | n/a | 0.894 a*–0.447 b*, b*, c* |
| **Refinement** | | | |
| Resolution (Å) | 36.99–1.94 | 46.9–2.5 | 47.82–3.62 |
| Reflections / free % | 57709/5.00 | 27532/5.00 | 11510/4.45 |
| $R_{work}$ / $R_{free}$ | 0.190/0.225 | 0.197/0.290 | 0.262/0.320 |
| B-average (Å$^2$) | 38.64 | 70.2 | 201.0 |
| R.m.s.d Bond lengths (Å) Bond angles (°) | 0.012 1.55 | 0.007 1.456 | 0.0071 1.6931 |
| Ramachandran (%) Favored Outliers | 98.6 0.0 | 93.7 0.0 | 87.1 1.0 |

Highest resolution shell is shown in parenthesis.
Data has been fitted to the ellipsoid defined by the following parameters:
Diffraction limits & principal axes of ellipsoid fitted to diffraction cut-off surface:
5.308 Å, 1.0000 0.0000 0.0000 0.894 a*–0.447 b*.
5.308 Å, 0.0000 1.0000 0.0000 b*.
3.240 Å, 0.0000 0.0000 1.0000 c*.
Worst diffraction limit after cut-off: 5.837 Å at reflection 20 3 0, in direction 0.989 a*+0.148 b*.
Best diffraction limit after cut-off: 3.623 Å at reflection 7 3 41, in direction 0.168 a*+0.072 b*+0.983 c*.
Beq: 366.94 [=equivalent overall isotropic B factor on Fs.].
Delta-B tensor: 100.25$^{B11}$ 100.25$^{B22}$ 200.49$^{B33}$.

(*Bricogne et al., 2017*). Crystallographic data statistics are presented in *Table 5*. The PDB deposition code for the SpNOX$_{DH}$ F397W is 8qq1.

## Data collection and structural solution of SpNOX$_{DH}$ WT

X-ray diffraction data were collected at ID30A-1/MASSIF-1 beamline at ESRF Grenoble (*Bowler et al., 2015*). A highly redundant data set at a resolution of 2.4 Å, including 3600 images with 0.1 degrees oscillation, was collected. The structure of SpNOX DH WT was solved at a 2.4 Å resolution by molecular replacement using the coordinates of the chain A of SpNOX DH F397W variant structure (PDB 8qq1) as a search template in Molrep searching for 3 mol/a.u. The initial solution was evaluated using COOT (*Emsley et al., 2010*) revealing no major clashes and good global packing with contacts between chains and classical size solvent channels. The final model was obtained by successive rounds of refinement using the programs REFMAC (*Murshudov et al., 1997*) and manual construction using COOT (*Emsley et al., 2010*). Crystallographic data statistics are presented in *Table 5*. The PDB deposition code for the SpNOX$_{DH}$ WT is 8qq5.

## Crystallization, data collection, and processing of full-length SpNOX

SpNOX F397W at 4.04 mg.ml$^{-1}$ in 50 mM TRIS pH 7, 300 mM NaCl, 0.025 mM LMNG, 0.01 mM FAD was crystallized in 30.5% PEG 300, 0.15 M Li$_2$SO$_4$, 0.15 M NaCl and 0.1 M MES pH 6, using the vapor diffusion method in a 6 µL protein sample plus 6 µL of well hanging drop at 20 °C. Hexagonal brick-red crystals were obtained, mounted on mesh litho-loops and flash frozen in native conditions in liquid nitrogen. X-ray diffraction data were collected at ID30A-1/MASSIF-1 beamline at ESRF Grenoble (*Bowler et al., 2015*). A dataset of 1800 images was collected with an oscillation range of 0.2° per image and an exposure time of 1.699 s and a crystal to detector distance refined to 500.9 mm. As observed for many membrane protein crystals (*Martin et al., 2023*; *Robert et al., 2017*), the data collected were very anisotropic going to 3.2 Å resolution in the strongest diffracting direction. After data processing, the dataset was finally cut at 3.6 Å to ensure sufficient completeness in the highest resolution shell. Inclusion of further data only resulted in more noise without obvious gain in 2mFo-DFc electron density. Data were initially processed using the program XDS as spherical to the highest resolution possible (3.2 Å) even though spherical statistics were not usable. Staraniso analysis for diffraction anisotropy (staraniso@globalphasing.org) revealed that completeness was 85.4% in the highest resolution shell, therefore revealing that all the data collectable for this crystal had been collected. Data was cut at the diffraction limits suggested by the Staraniso server using only the first 1000 images.

Phases were solved by molecular replacement using PHASER (*McCoy et al., 2007*), on amplitudes, with data corrected for anisotropy using Staraniso. The search models were the high-resolution DH domain (PDB 8qq5) and a model of the transmembrane domain obtained using PROMALS3D (*Pei et al., 2008*) and I-TASSER (*Yang and Zhang, 2015*) on the basis of multiple sequence alignment of NOX homologs and validated by structural alignment on CsNOX (*Vermot et al., 2020*). The molecular replacement solution was achieved with placing the DH domain first, followed by the TM domain, yielding LLG = 91.8 and TFZ = 10.9. The phasing resulted in a model with one molecule of SpNOX in the asymmetric unit and the space group P6$_4$22. Crystal packing generation showed a sensible solution (classic type II packing of membrane protein and no clash), showing good density for the DH domain and clearly identifiable transmembrane helices, representing a typical initial anisotropic low-resolution map for a membrane protein solved by molecular replacement.

The resulting model was not biologically possible with an obvious error in the positioning of the transmembrane domain, which was inverted by 180° around the longitudinal axis. A new transmembrane domain model was generated on the basis of a sequence alignment between CsNOX and SpNOX and the use of CHAINSAW (CCP4). A new molecular replacement with PHASER (*McCoy et al., 2007*) without modification of the DH domain position resulted in a new orientation of transmembrane domain, which was biologically compatible. Fo-Fc map of the model harbored two positive peaks at 6.1σ and 3.6 σ corresponding to the position of proximal and distal hemes, respectively, sandwiched between two key histidine residues, allowing the sequence assignment of the TM domain. Iterative manual building in COOT (*Emsley et al., 2010*) and LORESTR pipeline (*Kovalevskiy et al., 2016*), using TLS (5–183 for TM domain and 184–399 for DH domain) allowed large repositioning of some α–helixes of transmembrane domain. A final comparison of a SpNox model generated with AlphaFold (*Jumper et al., 2021*) drove the repositioning of some loops and some sequence shifts in transmembrane helices not associated with hemes. Final refinement was made with the help of ISOLDE (*Croll, 2018*) and LORESTR, ramachandran and rotamers outliers were corrected, yielding a final model with *R*=26.2% and Rfree = 32.0% that was deposited in the Protein Data Bank under the PDB accession code 8qq7. Crystallographic data statistics are presented in *Table 5*. The PDB deposition code for the SpNOX F397W is 8qq7.

## Acknowledgements

Funding for the automated crystallography pipelines at EMBL Grenoble was provided by the grants iNEXT Discovery (grant agreement ID: 871037) and Fragment Screen (grant agreement ID: 101094131) funded by the European Commission and through Instruct-ERIC. We want to thank the EMBL-ESRF Joint Structural Biology Group (JSBG) and in particular Andrew McCarthy for access and support in the use of MX beam lines at the ESRF. This work used the platforms of the Grenoble Instruct center (UMS 3518 CNRS-CEA-UJF-EMBL) with support from FRISBI (ANR-10- INSB-0502) and GRAL (ANR-10-LABX-49–01) within the Grenoble Partnership for Structural Biology. This work was supported by

the French Agence Nationale de la Recherche Bandit Project (ANR17-CE11-0013) to FF and JD. AV was supported through the Emergence program from the Univ. Grenoble Alpes; SMES was supported by an invited professorship from Univ. Grenoble-Alpes and through Emergence partner Kennesaw State University.

## Additional information

### Funding

| Funder | Grant reference number | Author |
|---|---|---|
| European Commission | iNEXT Discovery (grant agreement ID: 871037) | Jose Antonio Marquez |
| European Commission | Fragment Screen (grant agreement ID: 101094131) | Jose Antonio Marquez |
| Agence Nationale de la Recherche | FRISBI (ANR-10- INSB-0502) | Franck Fieschi |
| Agence Nationale de la Recherche | GRAL (ANR-10-LABX-49-01) | Franck Fieschi |
| Agence Nationale de la Recherche | Bandit (ANR17-CE11-0013) | Jerome Dupuy Franck Fieschi |
| Université Grenoble Alpes | Emergence program | Susan ME Smith Franck Fieschi |
| Université Grenoble Alpes | Invited professorship | Susan ME Smith |

The funders had no role in study design, data collection and interpretation, or the decision to submit the work for publication.

### Author contributions

Isabelle Petit-Hartlein, Annelise Vermot, Conceptualization, Validation, Investigation, Visualization, Writing – original draft; Michel Thepaut, Anne-Sophie Humm, Florine Dupeux, Data curation, Formal analysis, Investigation; Jerome Dupuy, Funding acquisition; Vincent Chaptal, Formal analysis, Methodology, Writing – review and editing; Jose Antonio Marquez, Formal analysis, Supervision, Methodology, Writing – review and editing; Susan ME Smith, Conceptualization, Formal analysis, Supervision, Visualization, Writing – original draft, Writing – review and editing; Franck Fieschi, Conceptualization, Formal analysis, Supervision, Funding acquisition, Visualization, Writing – original draft, Project administration, Writing – review and editing

### Author ORCIDs

Franck Fieschi ⬢ https://orcid.org/0000-0003-1194-8107

Reviewer #1 (Public Review): https://doi.org/10.7554/eLife.93759.3.sa1
Reviewer #2 (Public Review): https://doi.org/10.7554/eLife.93759.3.sa2
Author response https://doi.org/10.7554/eLife.93759.3.sa3

## Additional files

### Supplementary files

• MDAR checklist

### Data availability

Diffraction data have been deposited in PDB under the accession codes 8qq1, 8qq5 and 8qq7. All other data generated or analyzed during this study are included in the manuscript and figure supplements.

The following datasets were generated:

| Author(s) | Year | Dataset title | Dataset URL | Database and Identifier |
|---|---|---|---|---|
| Humm AS, Dupeux F, Vermot A, Petit-Harleim I, Fieschi F, Marquez JA | 2023 | SpNOX dehydrogenase domain, mutant F397W in complex with Flavin adenine dinucleotide (FAD) | https://www.rcsb.org/structure/8QQ1 | RCSB Protein Data Bank, 8QQ1 |
| Thepaut M, Petit-Hartlein I, Vermot A, Humm AS, Dupeux F, Marquez JA, Smith S, Fieschi F | 2023 | Structure of WT SpNox DH domain: a bacterial NADPH oxidase | https://www.rcsb.org/structure/8QQ5 | RCSB Protein Data Bank, 8QQ5 |
| Thepaut M, Petit-Hartlein I, Vermot A, Chaptal V, Humm AS, Dupeux F, Marquez JA, Smith S, Fieschi F | 2023 | Structure of SpNOX: a Bacterial NADPH oxidase | https://www.rcsb.org/structure/8QQ7 | RCSB Protein Data Bank, 8QQ7 |

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
