## [Editor Report · eLife assessment]

In this manuscript, the authors investigate the properties of prokaryotic NADPH oxidases (NOX) and discuss the implications for NOX regulation and function. The structure of the *S. pneumoniae* Nox protein is an **important** step forward in our understanding of procaryotic NOX enzymes and the characterization and interpretation are **convincing**. The results will be of interest to structural biologists as well as biochemists focusing on enzymatic functions.

---

## [Referee Report · Reviewer #1 (Public Review)]

Summary:

The manuscript describes the crystal structures of *Streptococcus pneumoniae* NOXs. Crystals were obtained for the wild-type and mutant dehydrogenase domain, as well as for the full-length protein comprising the membrane domain. The manuscript further carefully studies the enzyme's kinetics and substrate-specificity properties. Streptococcus pneumoniae NOX is a non-regulated enzyme, and therefore, its structure should provide a view of the NOX active conformation. The structural and biochemical data are discussed on this ground.

Strengths:

This is very solid work. The protein chemistry and biochemical analysis are well executed and carefully described. Similarly, the crystallography must be appreciated given the difficulty of obtaining good enzyme preparations and the flexibility of the protein. Even if solved at medium resolution, the crystal structure of the full-length protein conveys relevant information. The manuscript nicely shows that the domain rotations are unlikely to be the main mechanistic element of NOX regulation. It rather appears that the NADPH-binding conformation is pivotal to enzyme activation. The paper extensively refers to the previous literature and analyses the structures comprehensively with a comparison to previously reported structures of eukaryotic and prokaryotic NOXs.

---

## [Referee Report · Reviewer #2 (Public Review)]

The authors describe the structure of the *S. pneumoniae* Nox protein (SpNOX). This is a first. The relevance of it to the structure and function of eukaryotic Noxes is discussed in depth.

One of the strengths of this work is the effort put into preparing a pure and functionally active SpNOX preparation. The protein was expressed in *E. coli* and the purification and optimization of its thermostability and activity are described in detail, involving salt concentration, glycerol concentration, and pH.

Comments on revised version:

This reviewer would like to compliment the authors for the conscientious revision of the manuscript. Their response to the comments and the detailed explanations of the issues that did not appear clear enough to the reviewer are much appreciated. Their reaction to the review was not only superbly competent but also prominently good natured.

The revised version is perfect and represents a major contribution to our understanding of the molecular details of Nox function. As for the questions not yet answered, I shall quote the authors: "Time will tell".

---

## [Author Response]

The following is the authors’ response to the original reviews.

**Public Reviews:**

**Reviewer #1 (Public Review):**
Summary:The manuscript describes the crystal structures of Streptococcus pneumoniae NOXs. Crystals were obtained for the wild-type and mutant dehydrogenase domain, as well as for the full-length protein comprising the membrane domain. The manuscript further carefully studies the enzyme's kinetics and substrate-specificity properties. Streptococcus pneumoniae NOX is a non-regulated enzyme, and therefore, its structure should provide a view of the NOX active conformation. The structural and biochemical data are discussed on this ground.Strengths:This is very solid work. The protein chemistry and biochemical analysis are well executed and carefully described. Similarly, the crystallography must be appreciated given the difficulty of obtaining good enzyme preparations and the flexibility of the protein. Even if solved at medium resolution, the crystal structure of the full-length protein conveys relevant information. The manuscript nicely shows that the domain rotations are unlikely to be the main mechanistic element of NOX regulation. It rather appears that the NADPH-binding conformation is pivotal to enzyme activation. The paper extensively refers to the previous literature and analyses the structures comprehensively with a comparison to previously reported structures of eukaryotic and prokaryotic NOXs.

We thank the referee for these very nice comments about our work.

Weaknesses:The manuscript is not always very clear with regard to the analysis of NADPH binding. The last section describes a "crevice" featured by the NADPH-binding sites in NOXs. It remains unclear whether this element corresponds to the different conformations of the protein C-terminal residues or more extensive structural differences. This point must be clarified.

We agree with the referee that our terminology was not very clear. Responding to your comment helped us to improve our explanation: we have changed the text to emphasize the differences we observe in the distances between the FAD binding groove and the entire NADPH binding groove, which includes conserved NADPH-contacting motifs as well as the critical aromatic.

A second less convincing point concerns the nature of the electron acceptor. The manuscript states that this NOX might not physiologically act as a ROS producer. A question then immediately arises: Is this protein an iron reductase?Can the authors better discuss or provide more data about this point?

The referee has a legitimate point, which was also our first idea. In the initial work on SpNOX, where we discovered bacterial NOX enzymes (see Hajjar et al 2017 in mBio), we evaluated its possible role as an iron reductase. There we showed that SpNOX can reduce CytC directly; however, while some reduction of Fe3+-NTA complex (used classically in ferric reductase activity assay) occurred, this reduction was inhibitable by SOD and occurred indirectly by the superoxide produced, so therefore not a true iron reductase activity. This represents a mixed situation of direct and indirect reduction of an iron-containing acceptor that appears to preclude physiological iron reductase activity since it appears that the protein component of CytC allows it to interact with SpNOX. As these questions had been already addressed in a previous paper, we did not add anything here and we prefer to underline this possibility of another acceptor and to leave this question open for future works.

**Reviewer #2 (Public Review):**
The authors describe the structure of the S. pneumoniae Nox protein (SpNOX). This is a first. The relevance of it to the structure and function of eukaryotic Noxes is discussed in depth.Strengths and WeaknessesOne of the strengths of this work is the effort put into preparing a pure and functionally active SpNOX preparation. The protein was expressed in *E. coli* and the purification and optimization of its thermostability and activity are described in detail, involving salt concentration, glycerol concentration, and pH.This reviewer was surprised by the fact that the purification protocol in the eLife paper differs from those in the mBio and Biophys. J. papers by the absence of the detergent lauryl maltose neopentyl glycol (LMNG). LMNG is only present in the activity assay at a low concentration (0.003%; molar data should be given; by my calculation, this corresponds to 30 μM).

We regret this misunderstanding: our description was not clear enough. As the referee points out, in previous papers we purified the full length SpNOX with the detergent LMNG. In the current paper, we described only the protocol for SpNOX DH domain variant, a soluble cytoplasmic domain. We have now modified the text to clarify the difference between the purification of fulllength SpNOX variants, which were performed with detergent as cited in Vermot et al 2020, and the purification of DH domains, which are soluble and thus did not require detergent in the purification.

In light of the presence of lipids in cryo-EM-solved structures of DUOX and NOX2, it is surprising that the authors did not use reconstitution of the purified SpNOX in phospholipid (nanodisk?).The issue is made more complicated by the statement on p. 18 of "structures solved in detergent like ours" when no use of detergent in the solubilization and purification of SpNOX is mentioned in the Methods section (p. 21-22).

As stated above, detergent used to purify the full-length version of SpNOX. We did in fact perform some preliminary tests of reconstitution in nanodiscs. Different trials of negative staining studies showed heterogeneous size of SpNOX in nanodiscs and the initial images were not promising. Furthermore, in parallel, we had positive results in crystallography relatively quickly with protein in detergent. We thus focused on refining the crystals, which was a fairly long and mobilizing task; we decided to allocate time and resources to the promising avenue and did not further pursue nanodiscs.

We did not go in theCryo-EM direction because the small size of the protein was initially believed to be a significant barrier to successful Cryo-EM. Perhaps we could have pursued this avenue: while our manuscript here was submitted to eLife, another group deposited a preprint in BioRxiv using CryoEM to solve the structure of SpNOX (see comment below). This structure was solved in detergent so even in this CryEM structure there is no information on the potential roles of lipids as asked by the referee.

In this revised version, we have added a comment, in the last paragraph, in reference to the additional data available today thanks to the other structures generated by this other group (Murphy's group).

Can the authors provide information on whether *E. coli* BL21 is sufficiently equipped for the heme synthesis required for the expression of the TM domain of SpNOX. Was supplementation with δaminolevulinic acid used

The production of His-SpNox in *E. coli* C41(DE3) is without any δ-aminolevulinic acid supplementation. Supplementation was tested but no change was observed regarding the heme content (UV/Visible spectra) so we settled on the purification described by Vermot et al 2020. Initially, for the mBio paper (Haajar et al 2017), we performed heme titrations which gave stoichiometry between 1.35 to 1.5 heme/protein, indicating 2 hemes (these data were not shown). In the end in this work we observed two hemes in the crystal structure, thus confirming that *E.coli*, at least for this protein, did not need supplementation with δ-aminolevulinic acid .

The 3 papers on SpNOX present more than convincing evidence that SpNOX is a legitimate Nox that can serve as a legitimate model for eukaryotic Noxes (cyanide resistance, inhibition by DPI, absolute FAD dependence, and NADPH/NADH as the donor or electrons to FAD). It is also understood that the physiological role of SpNOX in S. pneumoniae is unknown and that the fact that it can reduce molecular oxygen may be an experimental situation that does not occur in vivo.I am, however, linguistically confused by the statement that "SpNOX requires "supplemental" FAD". Noxes have FAD bound non-covalently and this is the reason that, starting from the key finding of Babior on NOX2 back in 1977 to the present, FAD has to be added to in vitro systems to compensate for the loss of FAD in the course of the purification of the enzyme from natural sources or expression in a bacterial host. I wonder whether this makes FAD more of a cosubstrate than a prosthetic group unless what the authors intend to state is that SpNOX is not a genuine flavoprotein.

We believe there is some confusion between SpNOX – the full length transmembran protein -- and SpNOXDH -- the cytosolic domain only. The sentence pinpointed by the referee was in fact “The strict requirement of FAD addition for SpNOXDH activity suggests that the flavin behaves as a cosubstrate”. This statement was about the isolated cytosolic domain that does not contain the TM part of the protein.

We agree that in WT NOX enzymes (including SpNOX) FAD is held within the enzyme structure and thus can be considered, by definition, as a prosthetic group. This is supported by the nanomolar affinity for FAD of SpNOX. We did not intend to say that NOX and SpNOX are not genuine flavoproteins.

On the other hand, when isolated, the affinity of DH domain for flavins drops to the µM level. This µM level of affinity does not allow stable maintenance of the flavin in the active site as illustrated by the spectra of Figure 3. This is instead the typical affinity of a substrate or a co-substrate (similar to that of substrate NADPH) that can be exchangeable and diffuse in and out of the active site. The DH domain recognizes and reduces flavins but, as a consequence of its lower affinity, will release to its environment free reduced flavins. Thus the isolated DH behaves as a flavin reductase that uses flavin as substrate. Such enzymes have already been well described (and some of them are of the FNR family). Such enzymes, using flavin as substrate, typically have affinity for flavin in the µM range and share with the SpNOX DH binding properties centered on the isoalloxazine ring only.

We understand that, in the text, to switch from the SpNOX to the SpNOX DH and for FAD from a prosthetic group to a diffusible co-substrate can be confusing. So, to make it clearer, we modified the following sentences and added references to “some flavin reductases characterization” that could provide support for the reader.

“The strict requirement of FAD addition for SpNOXDH activity and its µM level of affinity suggests that the flavin behaves as a co-substrate rather than a prosthetic group. As an isolated domain, SpNOXDH may work as a flavin reductase enzyme (Gaudu et al, 1994; Fieschi et al 1995; Nivière et al 1996), ..”

We hope that it will help.

I am also puzzled by the statement that SpNOX "does not require the addition of Cyt c to sustain superoxide production". Researchers with a Cartesian background should differentiate between cause and effect. Cyt c serves merely as an electron acceptor from superoxide made by SpNOX but superoxide production and NADPH oxidation occur independently of the presence of added Cyt c.

Thanks to the referee for pointing out this poor wording. We agree and have amended the text to clarify what we originally meant. It is now:

“SpNOXDH requires supplemental FAD to sustain both superoxide production, which can be observed in the presence of Cyt c (Figure 2A), and NADPH oxidation, which can be observed in the absence of Cyt c (Figure 2B).”

The ability of the DH domain of SpNOX (SpNOXDH) to produce superoxide is surprising to this reviewer.The result is based on the inhibition of Cyt c reduction by added superoxide dismutase (SOD) by 40%. In all eukaryotic Noxes superoxide is produced by the one-electron reduction of molecular oxygen by electrons originating from the distal heme, having passed from reduced FAD via two hemes. The proposal that superoxide is generated by direct transfer of electrons from FAD to oxygen deserves a more in-depth discussion and relies too heavily on the inhibitory effect of SOD. A control experiment with inactivated SOD should have been done (SOD is notoriously heat resistant and inactivation might require autoclaving).

The initial reports of a NOX DH-domain-only construct (that of human Nox4) producing superoxide are cited in the text. Moreover, natural flavin reductases are known to produce superoxide due to the release of free reduced flavin in the medium.

As explain above, FAD in full length SpNox is a relay for the electrons from NADPH to heme and is internal to the protein and thus devoted to this specific task.

In the case of SpNOX DH, its flavin reductase behavior leads to the release in the medium of free reduced flavin as a nonspecific diffusible electron carrier. It has been already demonstrated that such free reduced flavin can efficiently reduce soluble O2 and be a source of superoxide.

This has been particularly well documented in (Gaudu et al, 1994. J.Biol.Chem). We have added this reference to the text (see the modified sentence in a reply, 2 comments above).

Furthermore, we want to point to the referee that the link between flavin and superoxide production here is not only based on the inhibition by SOD. When we added the flavin inhibitor DPI we observed no more superoxide production from the DH domain (Figure 2C). This supports the role of free-reduced flavin in both the production of superoxide and also part of direct cyt C reduction as observed.

An unasked and unanswered question is that, since under aerobic conditions, both direct Cyt c reduction (60%) and superoxide production (40%) occur, what are the electron paths responsible for the two phenomena occurring simultaneously?

We thank the referee for dedication to a clear understanding of the mechanism used by the SpNOXDH construct. It pushes us to develop a clear description of the mechanism at work here for the readers. Please find below a proposal mechanism describing the electron transfer from NAD(P)H to free flavin that can, as diffusible species, then reduce non-specifically either the O2 or the Cyt.C encountered.

However, it is important to remember that this is not physiological, and rather the result of using a DH domain isolated from the TM of SpNOX. Nonetheless, it shows that the DH domain is fully functional for NAD(P)H as well as the hydride transfer.

This reviewer had difficulty in following the argument that the fact that the kcat of SpNOX and SpNOXDH are similar supports the thesis that the rate of enzyme activation is dependent on hydride transfer from nicotinamide to FAD.

We have amended the text to clarify this point. If the reaction rate is not affected by the presence or absence of the hemes in the TM domain, this inevitably implies that the rate is NOT limited by the electron transfer to the heme, and ultimately to O2, from the FAD, and thus the hydride transfer step that oxidizes the FAD must be the rate limiting step.

The section dealing with mutating F397 is a key part of the paper. There is a proper reference to the work of the Karplus group on plant FNRs (Deng et al). However, later work, addressing comparison with NOX2, should be cited (Kean et al., FEBS J., 284, 3302-3319, 2017). Also, work from the Dinauer group on the minimal effect of mutating or deleting the C-terminal F570 in NOX2 on superoxide production should be cited (Zhen et al., J. Biol. Chem. 273, 6575-6581, 1998).

We thank the reviewer for pointing out our unintended omission of these important works; we have amended the text and added the citations.

It is not clear why mutating F397 to W (both residues having aromatic side chains) would stabilize FAD binding.

In a few words, trp’s double ring can establish larger and stronger vanderWaals contact with the isoalloxazine ring than the phe sidechain. Our discussion regarding this point is extensive in the structural section where we compare the structures with F and W in this position. At this time we do not think it is necessary to add anything to the text.

Also, what is meant by "locking the two subdomains of the DH domain"? What subdomains are meant?

The two subdomains are the NADPH-binding domain and the FAD-binding domain, which we define on p 11 (“SpNOXDH presents a typical fold of the FNR superfamily of reductase domain containing two sub-domains, the FAD-binding domain (FBD) and an NADPH-binding domain (NBD) “) and which are labeled in Fig. 4. By “locking” we meant to convey immobilizing them into a specific conformation; we have amended the text to clarify this point.

Methodological details on crystallization (p. 11) should be delegated to the Methodology section. How many readers are aware that SAD means "Single Wavelength Anomalous Diffraction" or know what is the role of sodium bromide?

We have amended the text to emphasize the intended point, which is the different origins of the two DH structures: the de novo structure was possible through co crystallization with bromide, and the molecular replacement structure used the de novo structure as a model.

The data on the structure of SpNOX are supportive of a model of Nox activation that is "dissident" relative to the models offered for DUOX and NOX2 activation. These latter models suggested that the movement of the DH domain versus the TM domain was related to conversion from the resting to the activated state. The findings reported in this paper show that, unexpectedly, the domain orientation in SpNOX (constitutively active!) is much closer to that of resting NOX2. One of the criteria associated with the activated state in Noxes was the reduction of the distance between FAD and the proximal heme. The authors report that, paradoxically, this distance is larger in the constitutively active SpNOX (9.2 Å) than that in resting state NOX2 (7.6 Å) and the distance in Ca2+-activated DUOX is even larger (10.2 Å).A point made by the authors is the questioning of the paradigm that activation of Noxes requires DH domain motion.Instead, the authors introduce the term "tensing", within the DH domain, from a "relaxed" to a more rigid conformation. I believe that this proposal requires a somewhat clearer elaboration

It is clear that the distance between the FAD and NADPH shown in the Duox and Nox2 structures is too large for the chemical reaction of hydride transfer. Wu et al used the terms ‘tense’ and ‘relaxed’ to describe conformations of the DH domain corresponding to ‘short distance’ and ‘longer distance’, respectively, between the two ligand binding sites. We quoted this terminology and have amended the text to clarify that we envision a motion of the NBD relative to the FBD, as distinct from a larger motion of the whole DH domain relative to the TM domain.

The statement on p. 18, in connection to the phospholipid environment of Noxes, that the structure of SpNOX was "solved in detergent" is puzzling since the method of SpNOX preparation and purification does not mention the use of a detergent. As mentioned before, this absence of detergent in the present report was surprising because LMNG was used in the methods described in the mBio and Biophys. J. papers. The only mention of LMNG in the present paper was as an addition at a concentration of 0.003% in the activity assay buffers.

Please see our response to similar points above. Detergent was present for the solubilization of the full-length SpNOX.

The Conclusions section contains a proposal for the mechanism of conversion of NOX2 from the resting to the activated state. The inclusion of this discussion is welcome but the structural information on the constitutively active SpNOX can, unfortunately, contribute little to solving this important problem. The work of the Lambeth group, back in 1999 (cited as Nisimoto et al.), on the role of p67-phox in regulating hydride transfer from NADPH to FAD in NOX2 may indeed turn out to have been prophetic. However, only solving the structure of the assembled NOX2 complex will provide the much-awaited answer. The heterodimerization of NOX2 with p22-phox, the regulation of NOX2 by four cytosolic components, and the still present uncertainty about whether p67-phox is indeed the final distal component that converts NOX2 to the activated state make this a formidable task.The work of the Fieschi group on SpNOX is important and relevant but the absence of external regulation, the absence of p22-phox, and the uncertainty about the target molecule make it a rather questionable model for eukaryotic Noxes. The information on the role of the C-terminal Phe is of special value although its extension to the mechanism of eukaryotic Nox activation proved, so far, to be elusive.

We really thank the referee for the positive comments on our work and the deep interest shown by this careful evaluation.

We understand the arguments of the referee regarding the relevance of our work here to eukaryotic NOX, but we do not share the reservations expressed. While human NOXes need interactions with other proteins or have EF-hand or other domains that control them, SpNOX corresponds exactly to the minimal core common to any NOX isoform. In fact, because SpNOX has only this conserved core, it is unique in that it can work as a constitutively active NOX without protein-protein interactions or regulatory domains. Thus the fundamentals of electron transfer mechanisms of NOX enzyme are present in SpNOX.

There might be some differences in the internal organization from isoform to isoform (as regarding the relative DH domain vs TM domain orientation) but considering the similarity between NOX2 and SpNOX topology we are rather confident that the SpNOX structure will turn out to be a reasonable model of the activated NOX2 structure. History will tell.

In any case, this work on SpNOX allowed us to highlight hydride transfer as the limiting step and also to highlight some structural differences that could be at the source of the regulation in eukaryotic NOX. In itself, we think this is a significant contribution to the field.

We warmly thank both referees for their constructive remarks and their help in the improvement of this manuscript.

**Recommendations for the authors:**

**Reviewer #1 (Recommendations For The Authors):**
The manuscript states that the flavin "behaves" like a co-substrate and thereby reports on the Km for the flavins. I feel that this terminology might be confusing. The flavin is unchanged after the reaction, and what matters is the enzyme's affinity for the flavin and the flavin concentration needed to saturate the enzyme (to have it in the fully holo form).

See above -- answering many questions from referee2, we have extensively commented on that point (substrate, cofactor, affinity, etc..) and made some adjustments in the text to clarify. We hope it is now satisfactory.

I could not find the methodological description of the experiments performed to measure the Km for the flavins, and the legend of Figure S4 does not help in this regard. I think that the data (left panels of S4) should be interpreted as binding curves with associated Kd values.

We have changed the text to clarify the method used to measure Km for flavins.

A related point is that the manuscript refers to Km as an "affinity". This is inappropriate and should be avoided, as the Km is not the Kd.

We agree with the referee that the Km is not the Kd. However, under the appropriate conditions, to which our experiments conform, Km is accepted as a relevant approximation of affinity (Srinisivan, FEBS Journal, v 289 pp 6086-6098 2022). We have added a sentence to clarify this point and cite this reference in the text.

The environment around the putative oxygen site should be shown. The text indicates that "the residues characteristic of the O2 reducing center in eukaryotic FRD domains of NOX and DUOX enzymes are not conserved in SpNOX." How does the site look? This point relates to the more general comment above on the oxidizing substrate used by this bacterial NOX.

This is a really interesting point that contains many potential biological developments for future studies of this prokaryotic family of NOX enzymes. While we were submitting this work to eLife for evaluation, another group (Murphy's lab) filed a pre-publication in BioRXiv, in which they also solved the structure of SpNOX but this time by CryoEM with an unexpected level of resolution for such a small protein (their paper is not yet published but probably under peer review somewhere). In their work, they made a special effort to identify the O2 reducing center (bacterial NOX sequences alignment, mutation studies, …) They were not able to localize such a site with accuracy. There is also other complementary data between their work and ours. So, we will add a paragraph at the end of the discussion to comment on this parallel work and to emphasize on the complementarity of their studies and what it brings to the final understanding of this enzyme.

The section "A Close-up View of NOX's NAD(P)H Binding Domains vs the FNR Gold Standard" should be clarified.

I found it difficult to understand. Is the different conformation of Phe397 creating the crevice? Could NADPH be modeled in NOX2 and DUOX in the same conformation observed in FNR and modeled in the bacterial NOX? Or would there be clashes, implying the necessity of larger conformational changes to bring the nicotinamide closer to the FAD?

Please see responses above on this point; we have amended the text to clarify. In a few words, we propose that activation in the eukaryotic enzymes would entail NBD subdomain (containing NADPH site) towards the FBD subdomain (containing FAD) through an internal motion within the DH domain. Doing so, they would approach the DH domain topology of SpNOX, which models an active state.

**Reviewer #2 (Recommendations For The Authors):**
On p. 6, second line, it should be (Figure 1C and 1D). Space is missing between C and "and".On p. 9, in Figure 3, the labeling A and B are missing. Also, the legend of part B does not correspond to the actual graph colors. Thus, the tracing of F397W is red and not grey as indicated in the legend.

Corrected. Thank you